# SON drives oncogenic RNA splicing in glioblastoma by regulating PTBP1/PTBP2 switching and RBFOX2 activity

Jung-Hyun Kim[1,2,8], Kyuho Jeong[2,8], Jianfeng Li[1,3,8], James M. Murphy [2], Lana Vukadin[4], Joshua K. Stone[1], Alexander Richard[1], Johnny Tran[1], G. Yancey Gillespie [5], Erik K. Flemington [6], Robert W. Sobol [1,3✉], Ssang-Teak Steve Lim [2✉] & Eun-Young Erin Ahn [4,7✉]

While dysregulation of RNA splicing has been recognized as an emerging target for cancer therapy, the functional significance of RNA splicing and individual splicing factors in brain tumors is poorly understood. Here, we identify SON as a master regulator that activates PTBP1-mediated oncogenic splicing while suppressing RBFOX2-mediated non-oncogenic neuronal splicing in glioblastoma multiforme (GBM). SON is overexpressed in GBM patients and SON knockdown causes failure in intron removal from the PTBP1 transcript, resulting in PTBP1 downregulation and inhibition of its downstream oncogenic splicing. Furthermore, SON forms a complex with hnRNP A2B1 and antagonizes RBFOX2, which leads to skipping of RBFOX2-targeted cassette exons, including the *PTBP2* neuronal exon. *SON* knockdown inhibits proliferation and clonogenicity of GBM cells in vitro and significantly suppresses tumor growth in orthotopic xenografts in vivo. Collectively, our study reveals that SON-mediated RNA splicing is a GBM vulnerability, implicating SON as a potential therapeutic target in brain tumors.

[1] Mitchell Cancer Institute, University of South Alabama, Mobile, AL, USA. [2] Department of Biochemistry and Molecular Biology, College of Medicine, University of South Alabama, Mobile, AL, USA. [3] Department of Pharmacology, College of Medicine, University of South Alabama, Mobile, AL, USA. [4] Department of Pathology, Division of Molecular and Cellular Pathology, University of Alabama at Birmingham, Birmingham, AL, USA. [5] Department of Neurosurgery, University of Alabama at Birmingham, Birmingham, AL, USA. [6] Department of Pathology, Tulane University School of Medicine, Tulane Cancer Center, New Orleans, LA, USA. [7] O'Neal Comprehensive Cancer Center, University of Alabama at Birmingham, Birmingham, AL, USA. [8] These authors contributed equally: Jung-Hyun Kim, Kyuho Jeong, Jianfeng Li. ✉email: rwsobol@southalabama.edu; stlim@southalabama.edu; eyahn@uabmc.edu

Glioblastoma multiforme (GBM) is the most common and lethal brain malignancy. Patients with GBM have a poor prognosis with a median survival of only one year after diagnosis, and <5% of patients survive more than 5 years[1–3]. Large-cohort molecular profiling studies identified that, in GBM patients, there are frequent genetic aberrations in a specific set of genes, such as *EGFR*, *CDKN2A/B*, *RB1*, *TP53*, *IDH1*, *NF1*, *PTEN*, and *PDGFRA*[4,5]. Our current understanding of the basis of GBM is largely limited to these several frequently mutated genes and their related pathways such as receptor tyrosine kinase, p53, and RB signaling pathways[6,7]. Nevertheless, GBM cells acquire broad and complex gene expression changes that cannot be explained solely by a few genetic mutations, suggesting there should be other factors contributing to abnormal gene expression in GBM. Therefore, identification of key GBM dependencies besides genetic mutations is urgently needed to develop novel therapeutic strategies for this deadly tumor.

Alternative RNA splicing is a critical mechanism for post-transcriptional gene regulation which contributes to proteomic diversity[8–10]. Accumulating evidence has demonstrated that aberrant RNA splicing due to splice site mutations and/or splicing factor mutations directs oncogenic gene expression in multiple types of solid tumors and hematologic malignancies[11,12]. Interestingly, the brain is one of the organs with the highest expression levels of RNA-binding proteins (RBPs)[13], implicating the importance of RNA regulation in the brain. In addition, neuronal differentiation accompanies extensive alternative splicing events to facilitate the inclusion of neuronal-specific cassette exons (neuronal exon)[14–16]. The most well-known regulatory event during neural differentiation is polypyrimidine tract-binding protein 1 (PTBP1) switching to its paralog PTBP2. PTBP1 is abundantly expressed in non-neuronal cells as well as undifferentiated neural stem cells, and binds to introns to induce neuronal exon skipping. During the onset of neuronal differentiation, PTBP1 is downregulated and in turn, PTBP2 which is repressed by PTBP1, is upregulated to increase neural exon inclusion and induce differentiation[17–19]. PTBP1 is highly expressed in GBM as well as astrocytoma, anaplastic astrocytoma, and medulloblastoma[20]. Recent studies further showed that upregulation of PTBP1 in GBM facilitates exon skipping events at PTBP1 target RNAs, resulting in the generation of oncogenic isoforms that promote cell proliferation and angiogenesis[21]. Besides PTBP1, several members of the heterogeneous nuclear ribonucleoprotein (hnRNP) family, which function in alternative RNA splicing, are also upregulated in GBM[22,23]. These findings clearly demonstrate that dysregulation of the RNA splicing program is indeed closely associated with GBM. However, detailed molecular mechanisms and crosstalk between splicing regulators in GBM are largely unknown.

SON is a nuclear speckle protein that contains both DNA- and RNA-binding domains and is particularly upregulated in embryonic stem cells and hematopoietic stem cells as well as leukemic blasts[24–27]. Our group previously identified SON as a co-factor that facilitates the splicing of weak splice sites[28]. Depletion of SON causes incomplete intron removal and alternative splicing, and many SON-targeted RNAs are particularly associated with cell cycle (especially mitotic progress) and DNA repair/replication[24,28,29]. We also recently reported that heterozygous loss-of-function mutations in the *SON* gene cause brain cortex malformation in humans, indicating that SON-mediated RNA splicing plays a critical role in gene expression in the brain[30].

Here, we show that SON is aberrantly upregulated in malignant brain tumors with the highest expression level in GBM, the most aggressive form of glioma, and there is a strong correlation between SON upregulation and short patient survival. We demonstrate that SON upregulates PTBP1 (oncogenic RBP) by facilitating intron removal from the *PTBP1* transcript, and suppresses the expression of PTBP2 (neuronal differentiation-promoting RBP) by controlling alternative splicing. Furthermore, we discovered that SON abrogates RBFOX2-mediated cassette exon inclusion during alternative splicing in cooperation with hnRNP A2B1. Our results reveal GBM cells depend on SON-mediated RNA splicing for their survival, clonogenicity, and tumorigenicity in vivo, implicating the therapeutic potential of SON inhibition in GBM.

## Results

**SON is upregulated in brain tumor patients, and high SON expression is correlated with short patient survival.** Previous studies by our group and others demonstrated SON's function in facilitating cell cycle progression and stemness as well as its potential significance in gene expression in the brain[24,28–30]. We therefore examined whether SON expression is altered in brain tumors. We performed reverse transcription and quantitative PCR (RT-qPCR) using several different primer sets (Fig. 1a) to measure SON and its isoform expression levels in our patient cohort of malignant brain tumors including anaplastic oligo-dendroglioma, anaplastic astrocytoma and GBM (Supplementary Table 1). Interestingly, *SON* transcripts are significantly upregulated in brain tumors compared to normal brain samples with the highest level observed in GBM patients (Fig. 1b). Unlike leukemia where we reported that the short isoforms of SON (SON E and SON B) accounts for most of the upregulated SON[27], we found that full-length SON (SON F) and its short isoforms are similarly upregulated in brain tumors. These results indicate that *SON* transcription is increased in brain tumors without changes in alternative splicing of *SON* transcripts.

We also analyzed SON expression using publicly available patient databases (REMBRANDT; Repository of Molecular Brain Neoplasia Data[31], R2 Database; http://r2.amc.nl, Oncomine[32], Exon Expression Array[33], GlioVis Data Portal; http://gliovis.bioinfo.cnio.es[34]) and corroborated our finding of SON upregulation in brain tumors (Fig. 1c and Supplementary Fig. 1a–c). We also found that SON expression levels are neither significantly different among GBM subtypes nor different G-CIMP (glioma cytosine-phosphate-guanine (CpG) island methylator phenotype) status (Supplementary Fig. 1d).

Importantly, Kaplan–Meier survival analysis obtained from the GlioVis Data Portal demonstrated that high levels of SON expression correlate with short survival of brain tumor patients. Negative correlations between SON level and patient survival were identified not only in high-grade GBM, but also in low-grade gliomas from multiple cohorts (REMBRANDT, TCGA and CGGA cohorts; visualized by GlioVis; Fig. 1d and Supplementary Fig. 2). Together, these data reveal significant upregulation of SON in malignant brain tumors and the potential role of SON overexpression in disease progression.

**Upregulation of SON is correlated with PTBP1 overexpression in GBM, and SON knockdown induces a switch in expression from PTBP1 to its paralog PTBP2.** To identify candidate genes regulated by SON in GBM, we sorted 307 genes whose expression levels are positively correlated with *SON* expression in two different databases of malignant gliomas (TCGA_GBM and Freije glioma data sets), but not in normal brain (Harris normal brain data set). Then, we also identified 386 genes that are significantly reduced upon *SON* knockdown in three independent studies with cell lines (Fig. 1e)[24,28,29]. Among these candidate genes, we found 13 genes commonly identified from all analyses. One of them was polypyrimidine tract-binding protein 1 (*PTBP1*) (Fig. 1e), which

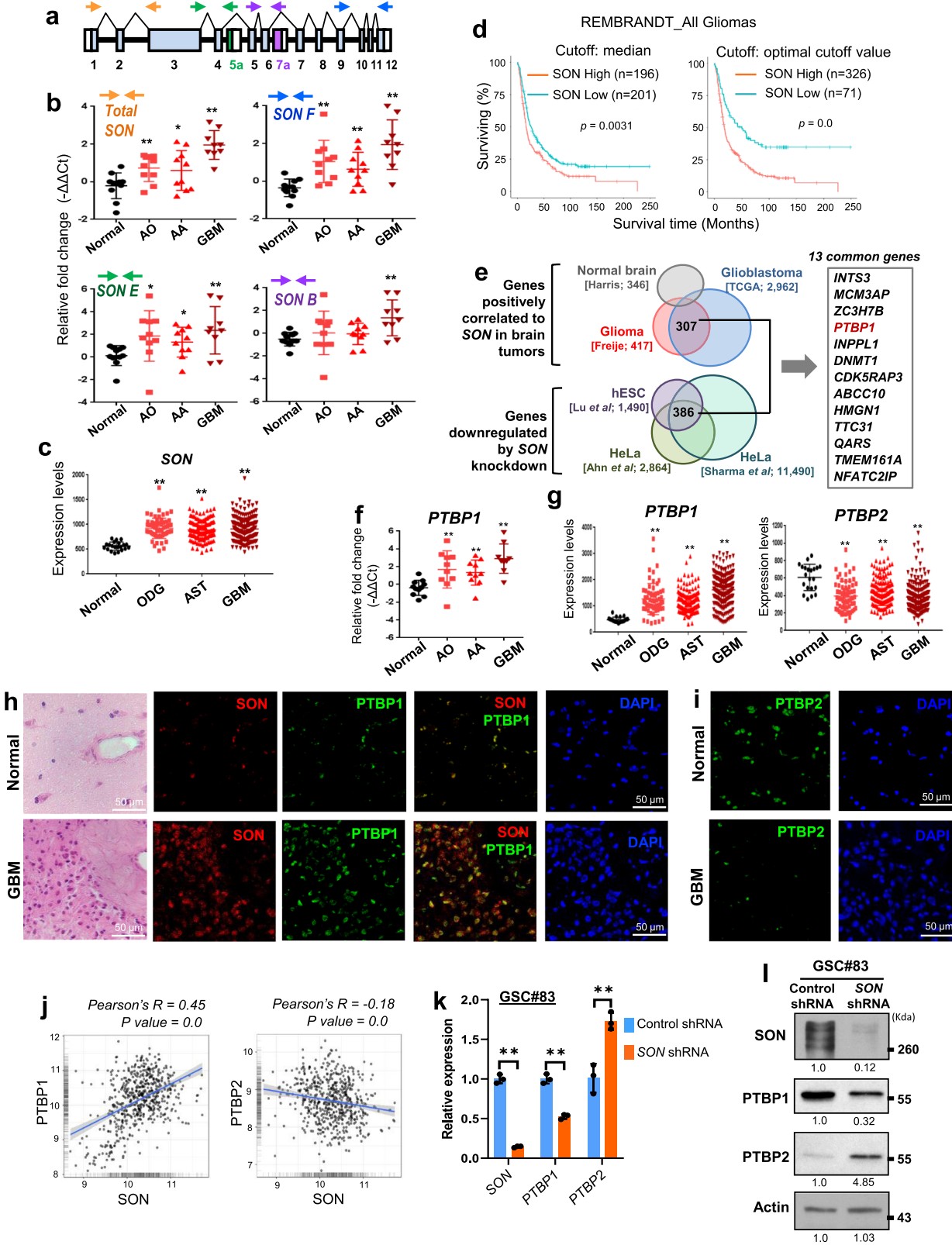

has been shown to be upregulated in several types of cancers[35–39] and to promote exon skipping, resulting in the expression of tumorigenic isoforms of several proteins (e.g., CD44, PKM2, PBX1, and ANXA7)[18,21,36–38,40–42]. Importantly, analysis of primary brain tumor samples from our patient cohort and from publicly available databases demonstrated that PTBP1 is abundantly expressed in brain tumors, particularly in GBM (Fig. 1f, g,

and Supplementary Fig. 3a, b). From immunostaining, we also observed significantly upregulated SON and PTBP1 in a GBM patient sample in which SOX2 expression, a well-known GBM stemness marker, was highly expressed (Fig. 1h and Supplementary Fig. 4).

Another member of the PTB protein family is PTBP2 (also called neuronal PTB, nPTB). While PTBP1 is a strong repressor

**Fig. 1 SON is overexpressed in malignant brain tumors and high SON levels correlate with short patient survival, upregulation of PTBP1 and downregulation of PTBP2. a** Schematic diagram of the *SON* gene. Boxes indicate twelve constitutive exons (E1–E12; light blue) and two alternative exons (E5a and E7a; green and purple, respectively). Color-coded arrows indicate qPCR primers used for detecting the common region present in all *SON* isoforms (orange arrows for total *SON*) or the specific regions present in each isoform (blue for *SON F*; green for *SON E*; purple for *SON B*). White boxes are 5′ or 3′ untranslated regions. **b** RT-qPCR analyses of *SON* isoform expression in normal brain and malignant brain tumor samples from our patient cohort. mRNA expression is standardized to *GAPDH*. Normal brain tissues, $n = 11$; AO anaplastic oligodendroglioma, $n = 10$; AA anaplastic astrocytoma, $n = 10$; GBM glioblastoma, $n = 9$. **c** Analyses of brain tumor data sets from REMBRANDT for total *SON* mRNA expression. Normal brain, $n = 21$; ODG oligodendroglioma, $n = 66$; AST, astrocytoma, $n = 145$; GBM, $n = 216$. **d** Kaplan–Meier overall survival curves of 397 glioma patients. High and low SON expression groups were divided by either the median value (left) or the optimal cutoff points determined by the GlioVis database (right). **e** Analyses of genes positively correlating with *SON*. **f** RT-qPCR analyses of *PTBP1* expression in our patient cohort. **g** Analyses of brain tumor data sets from REMBRANDT for the expression levels of *PTBP1* (left) and *PTBP2* (right) in ODG, AST, and GBM compared with normal brain. \*\*$p < 0.01$. **h** H&E staining and immunofluorescence images were obtained from human glioblastoma (GBM) patients (040201196; Supplementary Table 1) and normal brain (011201078; Supplementary Table 1) from our cohort. Representative images of SON (red), PTBP1 (green), and DAPI (blue) for GBM and normal brain were shown ($n = 3$). Scale bar: 50 μm. **i** Representative images of PTBP2 (green) and DAPI (blue) for GBM and normal brain were shown. Scale bar: 50 μm. **j** Correlation plots of *SON* vs. *PTBP1* (left) and *SON* vs. *PTBP2* (right) expression in REMBRANDT glioma patients ($n = 578$). Solid blue line indicates line of best fit, with shaded areas depicting standard deviation (SD) confidence intervals. **k, l** The effects of *SON* knockdown on the expression of PTBP1 and PTBP2 in GSC #83 were tested by RT-qPCR analysis (**k**) and Western blot analysis (**l**). Data from qPCR are presented as mean ± SD from three independent experiments. \*$p < 0.05$, \*\*$p < 0.01$. WB data are representative of $n = 3$ independent experiments. The number under each band in the Western blots indicates the average of relative band intensity from three independent WB experiments. Error bars in all graphs represent the standard deviation (SD) of tests. Statistical significance was determined by an unpaired two-tailed *t*-test. Source data are provided in the Source data file.

of neuronal-specific cassette exons (neuronal exons), PTBP2 is a weak repressor that allows neuronal exon inclusion. Therefore, a timely switch from PTBP1 to PTBP2 should occur during the early onset of neuronal differentiation[43–47]. Our database analyses as well as immunostaining of human GBM samples showed that PTBP2 expression is decreased in malignant glioma (Fig. 1g, i and Supplementary Fig. 3a, b). While PTBP1 expression has a strong positive correlation with SON expression, PTBP2 expression has a negative correlation with SON expression (Fig. 1j). Taken together, these findings demonstrate that SON overexpression in malignant gliomas, especially in GBM, has strong correlations with PTBP1 overexpression and PTBP2 downregulation, suggesting that there is a functional connectivity between SON, PTBP1, and PTBP2 in glioblastoma cells.

To further examine the role of SON in PTBP1 and PTBP2 expression, we reduced SON expression in the U87MG glioblastoma cell line and two different lines of patient-derived-glioma stem cells (GSCs), established previously (GSC #83, mesenchymal GSCs; GSC #84, proneural GSCs)[48], using validated siRNA or shRNA[27]. We found that PTBP1 is significantly downregulated upon *SON* knockdown in U87MG cells and GSCs, while PTBP2 expression is remarkably increased upon *SON* knockdown (Fig. 1k, l and Supplementary Fig. 3c, d). These results indicate that SON is a potential upstream regulator of the PTBP1/PTBP2 switch.

**SON knockdown causes *PTBP1* intron retention, resulting in decreased PTBP1 expression and suppression of the PTBP1-mediated splicing program.** We next sought to address how SON promotes PTBP1 expression in GBM. Our group and others have reported that SON functions as a splicing co-factor supporting correct processing of weak splice sites[28–30]. Close examination of all splice sites present in *PTBP1* followed by RT-PCR analyses demonstrated that *SON* knockdown causes incomplete removal of introns 4, 5, and 6 in *PTBP1* pre-mRNA (Fig. 2a and Supplementary Fig. 3e), revealing that PTBP1 expression is regulated by SON at the RNA splicing level.

Recent studies have demonstrated that there are many internal introns that are not efficiently removed during co-transcriptional splicing, and are thus present abundantly within polyadenylated transcripts[49]. Retention of these introns is not due to random splicing errors and it plays critical role in controlling gene expression. These introns are referred to as detained introns, and

the transcripts containing detained introns remain in the nucleus and are not subject to nonsense-mediated mRNA decay (NMD)[49–51]. Interestingly, we found that intron-retained *PTBP1* transcript is mainly present in the nucleus (Fig. 2b, c). In addition, we found that treatment with Emetine, an NMD inhibitor, does not cause any accumulation of the intron-retained *PTBP1* transcript (Fig. 2d) or total *PTBP1* transcript level (Fig. 2e), confirming that *PTBP1* transcript is not regulated by NMD upon *SON* knockdown. These findings indicate that the PTBP1 transcript contains detained introns that are not efficiently removed during co-transcriptional RNA splicing where it likely functions as a rate-limiting factor for the production of fully spliced mRNA and we found that SON plays a key role in removing this detained intron.

Importantly, we analyzed the samples from our patient cohort, and found that the ratio of intron-retained/total *PTBP1* is significantly lower in malignant glioma patient samples compared to normal brain samples (Fig. 2f). This result reveals that the introns from the *PTBP1* transcript is indeed more efficiently removed in malignant glioma compared to normal brain.

One of the mechanisms by which PTBP1 maintains stemness and non-neuronal properties in embryonic stem cells is via its regulation of pre-B cell leukemic homeobox 1 (PBX1)[18]. In early embryonic tissues, PTBP1 suppresses the inclusion of exon 7 of *PBX1*, generating the short isoform PBX1b. During neuronal differentiation, PTBP1 is downregulated in order to promote *PBX1* exon 7 inclusion and induce the isoform switch from PBX1b to PBX1a which promotes neuronal differentiation[18]. Analysis of both our glioma patient cohort and the database[33] confirmed the low level of *PBX1* exon 7 usage in GBM patients (Fig. 2g), indicating PTBP1-mediated exon skipping is a predominant event in malignant glioma. We also found that *SON* knockdown significantly increases PBX1a (exon 7-included form) in the U87MG cell line as well as two patient-derived GSC lines (Fig. 2h), indicating PTBP1-mediated exon skipping is efficiently blocked by *SON* knockdown.

We also examined the splicing of Reticulin 4 (*RTN4* gene; Nogo protein) as a target of PTBP1-mediated oncogenic splicing. It has been known that PTBP1 induces skipping of *RTN4* exon 3, generating the transcript (*RTN4-B*) for the Nogo-B protein which promotes tumor angiogenesis, epithelial-mesenchymal transition, cell migration, and proliferation[52]. When PTBP1 is reduced, the inclusion of exon 3 of *RTN4* is enhanced, subsequently generating

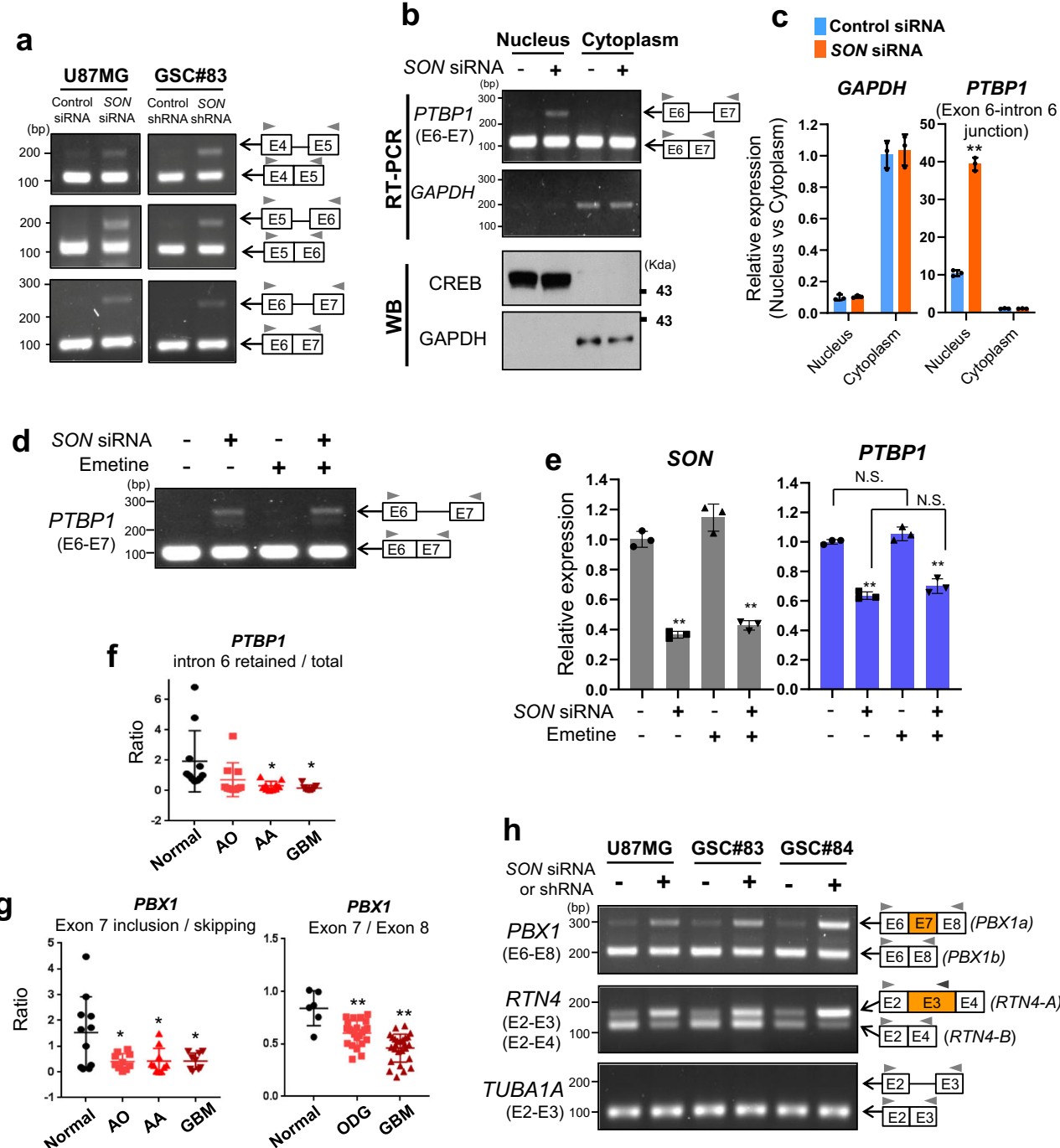

the transcript (*RTN4-A*) for the Nogo-A protein which blocks neuronal angiogenesis[52]. We observed that *SON* knockdown significantly increases exon 3 inclusion during *RTN4* splicing (Fig. 2h) and upregulates the Nogo-A protein (Supplementary Fig. 3d). Collectively, these data demonstrate that PTBP1-mediated exon skipping events, which generate splice products closely associated with poor differentiation and tumor progression, can be efficiently attenuated by SON knockdown.

**SON knockdown facilitates skipping of the *PTBP2* cassette exon (exon 10) prior to PTBP1 downregulation**. It has been well documented that PTBP1 is a central regulator (repressor) of PTBP2 expression[17]. In neural stem cells and undifferentiated precursors, PTBP1 expression is kept high which, in turn, promotes the skipping of a *PTBP2* neuronal cassette exon (exon 10),

which subsequently generates transcripts bearing a premature termination codon and are being subject to NMD[44]. We examined whether PTBP2 upregulation is due to enhanced *PTBP2* cassette exon inclusion and found that *SON* knockdown indeed markedly increased the exon 10-included form of the *PTBP2* transcript (Fig. 3a). Interestingly, a time-course analysis of gene expression revealed that while a more than two-fold increase of the *PTBP2* mRNA level was already detectable by 24 h after *SON* siRNA transfection, the *PTBP1* levels were not significantly decreased until 48 h after *SON* siRNA transfection (Fig. 3b). Further examination of the protein levels at earlier time points after *SON* siRNA transfection demonstrated that the increase of PTBP2 indeed occurred despite no significant change in PTBP1 level (Fig. 3c). The RT-qPCR experiments using specific exon-exon junction primers for calculation of a ratio between two

**Fig. 2 SON knockdown causes intron retention in the PTBP1 transcript, resulting in reduction of PTBP1 expression and inhibition of its downstream oncogenic RNA splicing. a** RT-PCR detecting *PTBP1* intron 4, intron 5, and intron 6 retention upon *SON* knockdown. Data are representative of $n = 3$ independent experiments. **b, c** RT-PCR (**b**) and RT-qPCR (**c**, $n = 3$) analyses demonstrate nuclear localization of intron 6 retained-*PTBP1* transcript. *GAPDH* mRNA was measured as a control for the cytoplasm-localized mRNA. Western blot analysis was performed using CREB (a nuclear marker) and GAPDH (a cytoplasmic marker) to assess fractionation quality was performed. RT-PCR and WB Data are representative of $n = 3$ independent experiments. **d** RT-PCR was performed to detect intron 6-retained *PTBP1* transcript from control or *SON* siRNA-transfected U87MG cells treated with 50 μM Emetine (+), an inhibitor of nonsense-mediated mRNA decay (NMD), or DMSO (−) for 3 h. *TUBA1A* exon 2–3 splicing was analyzed as a negative control. Data are representative of $n = 3$ independent experiments. **e** *SON* and *PTBP1* expression in control or *SON* siRNA-transfected U87MG cells treated with 50 uM Emetine (+) or DMSO (−) for 3 h. All values in the graphs are presented as mean ± SD. $n = 3$. NS; not significant, **$p < 0.01$. **f** Intron 6-retained *PTBP1* in our patient cohort was measured by qPCR using a forward primer targeting exon 6 and a reverse primer targeting the intron 6. Then, the ratio of the intron 6-retained form and total *PTBP1* was calculated. (Normal, $n = 11$; AO, $n = 10$; AA, $n = 10$; GBM, $n = 9$). **g** *PBX1* exon 7 inclusion/skipping ratio was calculated from qPCR using a forward primer targeting exon 5 and two different reverse primers targeting the exon 6–exon 7 junction or the exon 6–exon 8 junction (left) (Normal, $n = 11$; AO, $n = 10$; AA, $n = 10$; GBM, $n = 9$). *PBX1* exon usage was calculated by expression ratio between exon 7 (an exon skipped by PTBP1 action) and exon 8 (constitutive exon) from the Exon Expression Arrays dataset (right) (Normal, $n = 6$; ODG, $n = 23$; GBM, $n = 26$). **h** RT-PCR assays demonstrating the effect of *SON* knockdown on PTBP1-mediated alternative splicing of *PBX1* and *RTN4*. The exon 7-skipped form of *PBX1* (*PBX1b*) and the exon 3-skipped form of *RTN4* (*RTN4-B*; for Nogo-B protein) are oncogenic forms. Cassette exons that are skipped by PTBP1 action were indicated in orange. *TUBA1A* splicing was analyzed as a negative control. Gray arrow heads above the exons indicate the primers used for PCR. Data are representative of $n = 3$ independent experiments. Error bars in all graphs represent the standard deviation (SD) of tests. *$p < 0.05$, **$p < 0.01$. Statistical significance was determined by an unpaired two-tailed *t*-test. Source data are provided in the Source data file.

different types of *PTBP2* pre-mRNAs (*exon* 10-included or -skipped forms; Fig. 3d) further confirmed that the increase of *PTBP2* exon 10 inclusion preceded *PTBP1* downregulation upon *SON* knockdown (Fig. 3e).

To address whether the reduction of PTBP1, a known repressor of PTBP2 expression, upon *SON* knockdown has a causative role in upregulating PTBP2, we restored the PTBP1 level through ectopic PTBP1 expression in SON-depleted cells (Supplementary Fig. 5). To our surprise, overexpression of PTBP1 could not repress the early induction of PTBP2 expression upon *SON* knockdown (48 h time point) (Fig. 3f, g). We also found that *PTBP2* exon 10 inclusion is prominent upon SON depletion, regardless of exogenous PTBP1 expression (Fig. 3h and Supplementary Fig. 5). In contrast, *SON* siRNA-induced exon inclusion events within other PTBP1 targets, *PBX1* and *RTN4*, were completely or partially abrogated by ectopic PTBP1 expression (Fig. 3h), indicating SON regulation of *PBX1* and *RTN4* splicing is through PTBP1 action. These findings led us to hypothesize that although the reduction of PTBP1 still contributes to PTBP2 upregulation at later time points (such as 48 and 72 h time points in Fig. 3b), the early and sharp increase of PTBP2 upon *SON* knockdown is not due to PTBP1 down-regulation. Therefore, it is likely that SON employs additional mechanisms to repress *PTBP2* exon 10 in a PTBP1-independent manner.

**Inclusion of *PTBP2* exon 10 upon *SON* knockdown requires the expression of RBFOX2.** Our finding of SON-mediated cassette exon (exon 10) suppression during *PTBP2* alternative splicing was contradictory to the previous observations by our groups and others where SON is required for induction of exon inclusion in many transcripts. Therefore, we hypothesized that there should be other mechanisms by which SON regulates alternative splicing of *PTBP2* exon 10, besides facilitating recognition and processing of weak splice sites. A recent report on iCLIP-seq (individual nucleotide-resolution cross-linking immunoprecipitation coupled to high-throughput sequencing) identified that RBFOX2, a member of the RBFOX RNA-binding protein family, regulates a large number of alternative splicing events. It has been shown that RBFOX2 binds to its specific motifs [(U)GCAUG] present at the upstream or downstream regions from the cassette exons, and induces exon skipping (when it binds to the upstream intron) or exon inclusion (when binds to the downstream intron)[53–55]. Interestingly, a previous study showed that RBFOX2 binds to the

*PTBP2* transcript and that *RBFOX2* knockdown increases *PTBP2* exon 10 skipping[53]. Therefore, we examined whether PTBP2 regulation by SON is associated with RBFOX2's action. Surprisingly, *SON* knockdown failed to upregulate PTBP2 when *RBFOX2* was depleted (Fig. 4a, b). In line with this result, RBFOX2 overexpression augmented *SON* knockdown-mediated PTBP2 upregulation (Supplementary Fig. 6a). Furthermore, the inclusion of *PTBP2* exon 10 upon *SON* knockdown was diminished when *RBFOX2* was depleted (Fig. 4c), demonstrating that *SON* knockdown-induced *PTBP2* alternative splicing (exon 10 inclusion) requires RBFOX2 expression. In contrast, *SON* knockdown-induced downregulation of PTBP1 as well as PTBP1-target splicing (e.g., *PBX1* exon 7 splicing) was not affected by *RBFOX2* knockdown (Fig. 4a–c), indicating that RBFOX2 function is specifically associated with SON-mediated PTBP2 regulation, but not PTBP1 regulation.

**SON and RBFOX2 competitively bind to the intron sequences flanking the *PTBP2* cassette exon with the highest binding to the intron downstream of the cassette exon.** Since we found that RBFOX2 expression is required for *SON* knockdown-mediated *PTBP2* exon 10 inclusion, we next investigated whether both SON and RBFOX2 are associated with the *PTBP2* transcript. We performed CLIP followed by qPCR (CLIP-qPCR) and demonstrated that SON indeed interacts with *PTBP2* pre-mRNA (Fig. 4d). SON binding was especially enriched near the intron-exon and exon-intron boundaries of exon 10 (the regions targeted by the primer sets 5 and 6; Fig. 4e, f) as well as the downstream intron region containing consensus RBFOX2 binding motifs (the region targeted by the primer set 7; Fig. 4e, f). Interestingly, RBFOX2-showed a similar enrichment pattern as SON on *PTBP2* transcripts and the highest RBFOX2-binding was observed at the region with RBFOX2-binding motifs present downstream of exon 10 (the region targeted by the primer set 7; Fig. 4e and blue bars of Fig. 4h). Since SON- and RBFOX2-binding sites largely overlap, we next questioned whether SON and RBFOX2 affect each other's RNA-binding ability. Interestingly, our CLIP-qPCR experiments demonstrated that *RBFOX2* knockdown greatly enhances SON-binding to *PTBP2* RNA (Fig. 4g). In line with this result, *SON* knockdown enhanced RBFOX2-binding to the target RNA (Fig. 4h). These results demonstrate that SON and RBFOX2 mutually suppress each other for interaction with *PTBP2* mRNA at the exon10-flanking regions where they exert opposite functions in the inclusion or skipping of exon 10.

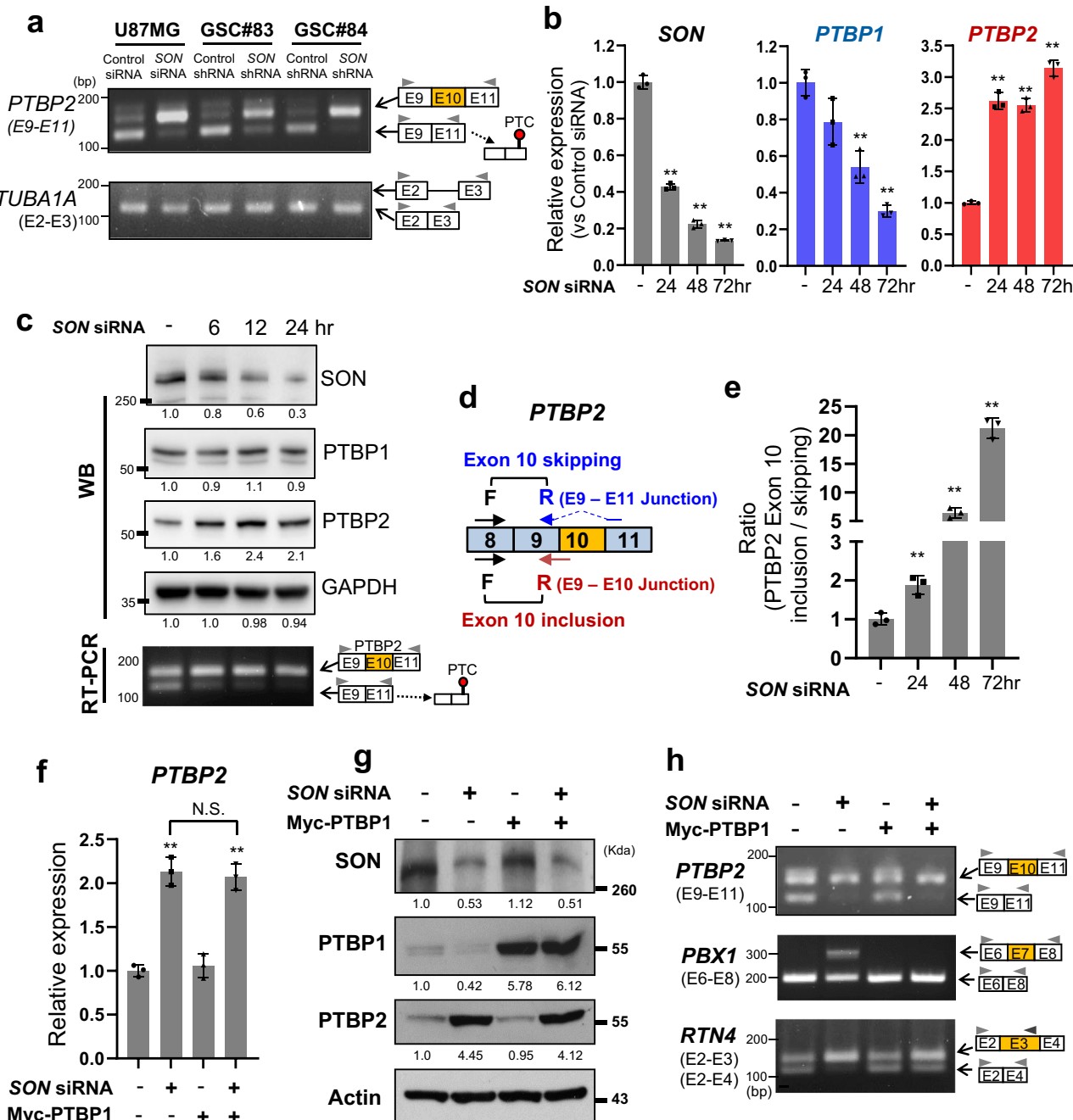

**Fig. 3 Knockdown of *SON* increases exon 10 inclusion in *PTBP2* transcript via PTBP1-independent mechanisms. a** RT-PCR analyses of *PTBP2* exon 10 inclusion upon *SON* knockdown in the U87MG cell line and two patient-derived GSC lines (GSC #83 and GSC #84). PTBP2 exon 10 skipping cases frameshifting, resulting in a premature termination codon (PTC; indicated with a red circle). Data are representative of $n = 3$ independent experiments. **b** *SON*, *PTBP1,* and *PTBP2* mRNA expressions at the indicated time points after *SON* siRNA transfection were analyzed by RT-qPCR. *GAPDH* was used for normalization ($n = 3$). **c** SON, PTBP1, and PTBP2 expression after *SON* siRNA transfection in U87MG cells were analyzed by Western blotting at the indicated time points. GAPDH was used as a loading control. The inclusion of *PTBP2* exon 10 was analyzed by RT-PCR upon SON knockdown. Data are representative of $n = 3$ independent experiments. **d** A schematic illustrating the qPCR strategy used to measure *PTBP2* exon 10 inclusion and skipping. A specific reverse primer detecting exon 9-11 junction (top; for exon 10 skipping) and exon 9–10 junction (bottom; for exon 10 inclusion) were used for qPCR. **e** The ratio of *PTBP2* exon 10 inclusion/exon 10 skipping determined by qPCR analysis using the primer sets shown in panel (**d**) ($n = 3$). **f, g** RT-qPCR (**f**, $n = 3$) and Western blot (**g**) assays demonstrate that *SON* knockdown-mediated-PTBP2 upregulation is through a PTBP1-independent mechanism. U87MG cells were transfected with *SON* siRNA and/or Myc-PTBP1 constructs as indicated and harvested in 48 h. WB Data are representative of $n = 3$ independent experiments. **h** RT-PCR analysis of *PTBP2* exon 10, *PBX1* exon 7, and *RTN4* exon 3 usages (skipping or inclusion) under the condition described for (**f**) and (**g**). Data are representative of $n = 3$ independent experiments. Error bars in all graphs represent the standard deviation (SD) of tests. NS; not significant, *$p < 0.05$, **$p < 0.01$. Statistical significance was determined by an unpaired two-tailed *t*-test. The number under each band in the Western blots indicates relative band intensity. Source data are provided in the Source data file.

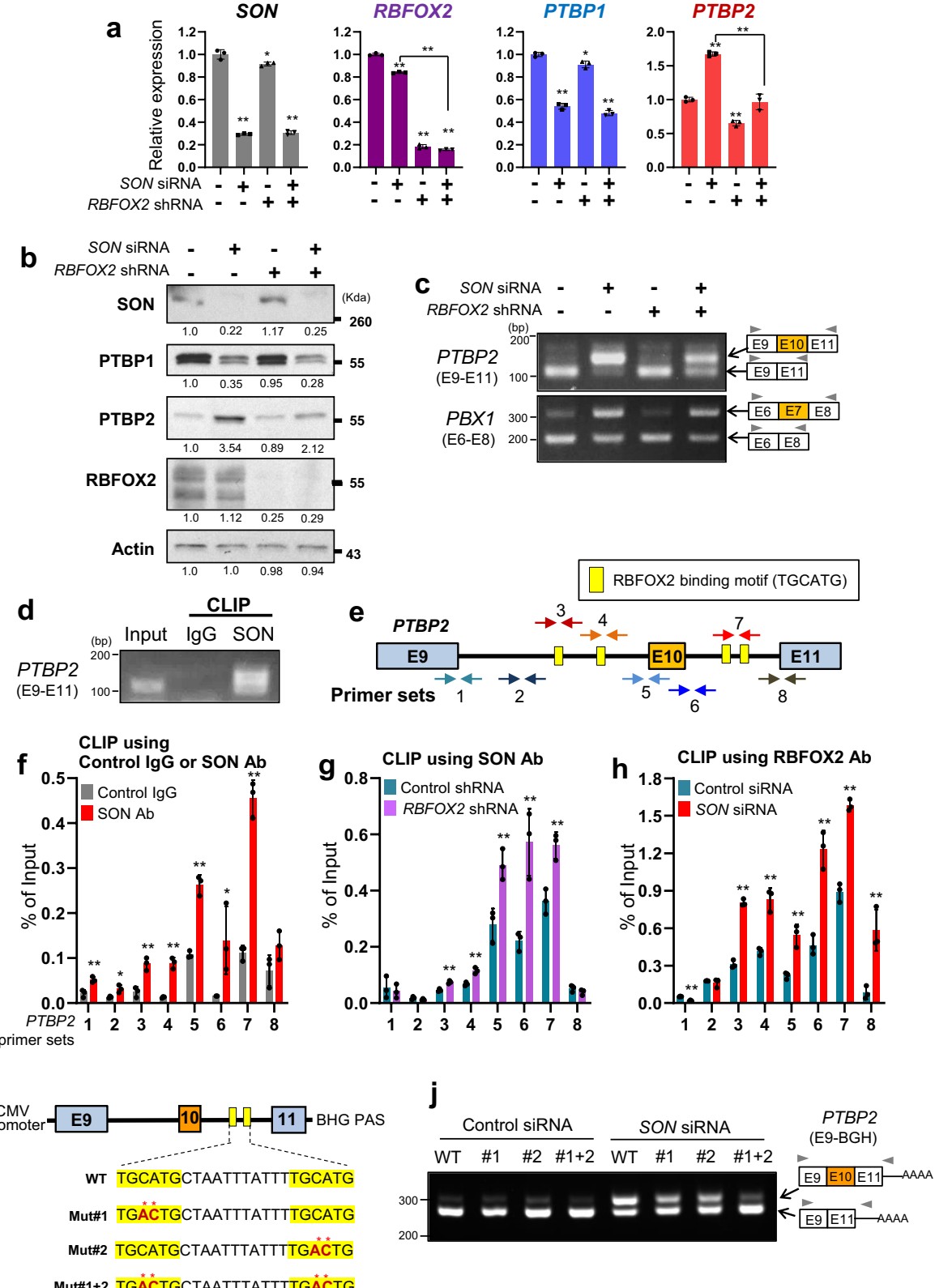

To further corroborate the importance of RBFOX2-binding sites in cassette exon inclusion upon *SON* knockdown, we used a *PTBP2* minigene covering the sequence from exon 9 to exon 11 (Supplementary Fig. 6b). Two RBFOX2-binding motifs are present in the intron downstream of the cassette exon, and we altered these two RBFOX2-binding motifs individually (Mut#1 or Mut#2) and then together (Mut#1 + 2) (Fig. 4i). The minigene

splicing assays in U87MG cells confirmed that, indeed, exon 10 is largely skipped in all the mini-gene constructs tested. When SON was reduced by siRNA (Supplementary Fig. 6c), we observed exon 10 inclusion is markedly enhanced in the wild-type minigene, but the level of inclusion is decreased in the minigenes with one of the RBFOX2-binding motifs mutated (Mut#1 and Mut#2 minigenes). Notably, when both of RBFOX2-binding

**Fig. 4 RBFOX2 expression is required for the *SON* knockdown-mediated *PTBP2* exon 10 inclusion, and RBFOX2 and SON compete with each other for binding to the target RNA. a** RT-qPCR analyses of *PTBP1* and *PTBP2* expression in U87MG cells after knockdown of *SON* and/or *RBFOX2* as indicated ($n = 3$). **b** The protein levels of SON, PTBP1, PTBP2, and RBFOX2 after knockdown of *SON* and/or *RBFOX2* were analyzed by Western blotting. Actin was used as a loading control. WB Data are representative of $n = 3$ independent experiments. **c** RT-PCR analyses of *PTBP2* exon 10 inclusion upon *SON* knockdown and/or *RBFOX2* knockdown. *PBX1* alternative splicing was shown as a control that is not regulated by RBFOX2. Data are representative of $n = 3$ independent experiments. **d** CLIP-PCR assay demonstrating the direct interaction between the SON protein and *PTBP2* RNA (exon 9 to 11 region). Data are representative of $n = 3$ independent experiments. **e** A diagram of the human *PTBP2* pre-mRNA with four RBFOX2-binding motifs (yellow boxes) and the locations of the primer sets (color-coded arrows designated with the numbers 1–8) that were used for the CLIP-qPCR assays shown in (**f–h**). **f** SON-binding to *PTBP2* pre-mRNA was measured by CLIP with a SON antibody followed by qPCR using the primer sets indicated by the numbers 1–8 ($n = 3$). Control IgG was used for a CLIP control. The bar graph shows qPCR signals amplified from the CLIP assays as percentage of amplification of the input RNA. **g** The effect of *RBFOX2* knockdown on SON-binding to *PTBP2* pre-mRNA was measured by CLIP-qPCR assays with U87MG cells expressing control shRNA and *RBFOX2* shRNA ($n = 3$). **h** The effect of *SON* knockdown on RBFOX2-binding to *PTBP2* pre-mRNA was measured by CLIP-qPCR assays with U87MG cells transfected with control siRNA and *SON* siRNA ($n = 3$). The experiments were performed three times **i** *PTBP2* minigenes containing the genomic DNA sequence from *PTBP2* exon 9 to exon 11 downstream of the CMV promoter. Minigenes with wild-type (WT) or mutated (Mut) RBFOX2 binding motifs (yellow boxes) were cloned. Red asterisks indicate the mutated nucleotides. **j** RT-PCR analyses of exon 10 inclusion/skipping using WT and RBFOX2-binding site-mutated minigenes in control or *SON* siRNA-transfected U87MG cells. Data are representative of $n = 3$ independent experiments. Error bars in all graphs represent the standard deviation (SD) of tests. NS not significant, *$p < 0.05$, **$p < 0.01$. Statistical significance was determined by an unpaired two-tailed *t*-test. The number under each band in the Western blots indicates relative band intensity. Source data are provided in the Source data file.

motifs were mutated, SON knockdown-induced exon 10 inclusion was largely abrogated (Fig. 4j). These results confirmed the key role of RBFOX2-binding to its consensus motifs in achieving cassette exon inclusion upon *SON* knockdown.

**SON and RBFOX2 form unique hnRNP complexes and hnRNP A2B1, a SON complex-specific component, cooperates with SON to antagonize RBFOX2-mediated *PTBP2* cassette exon inclusion.** A recent paper showed that RBFOX2 is associated with several RBPs, such as hnRNPs and helicases, and forms the LASR (Large Assembly of Splicing Regulators) complex to regulate target RNA splicing[56]. To dissect the underlying mechanisms of SON and RBFOX2 function in oppositely regulating cassette exon inclusion/skipping, we next questioned whether SON also functions in the context of a protein complex. Therefore, we performed immunoaffinity purification and mass-spectrometry to identify proteins that are associated with SON (Supplementary Data 1). We found many U2 snRNP components, U2-associated proteins, and hnRNP family proteins in the pull-down sample with a specific SON antibody (Supplementary Fig. 7 and Supplementary Data 1). To verify SON's interactions with hnRNPs, we performed immunoprecipitation (IP) and Western blotting (WB) using U87MG GBM cells. The results confirmed that SON indeed interacts with hnRNP M, hnRNP H, hnRNP K, and hnRNP A2B1, but does not interact with RBFOX2 and hnRNP U (Fig. 5a, b). In contrast, RBFOX2 interacts with hnRNP M, hnRNP H, and hnRNP U (Fig. 5a, b), which is consistent with the previously reported LASR complex. These findings indicate that SON and RBFOX2 are not present in the same protein complex, and both SON and RBFOX2 complexes contain hnRNP H and hnRNP M while hnRNP K and hnRNP A2B1 are present only in the SON complex.

To examine whether SON-interacting hnRNPs play a role in PTBP2 expression, we performed knockdown experiments using specific shRNAs for each hnRNP. Interestingly, depletion of hnRNP A2B1 significantly increased PTBP2, recapitulating *SON* knockdown, while the depletion of *hnRNP K* or *hnRNP M* did not alter PTBP2 level (Fig. 5c). Although it has been shown that hnRNP H is overexpressed in glioma and drives oncogenic splicing[22], knockdown of *hnRNP H* did not alter PTBP1 or PBTP2 levels (Supplementary Fig. 8). Furthermore, *SON* and *hnRNP A2B1* double knockdown led to a higher increase of PTBP2 expression as well as *PTBP2* exon 10 inclusion compared to single knockdown of *SON* or *hnRNP A2B1* (Fig. 5d, e). We also detected co-localization of SON and hnRNP A2B1 in nuclear

speckles, but the localization of these two proteins was not completely identical (Fig. 5f). This result supports the notion that SON and hnRNP A2B1 function cooperatively as well as independently.

In an analysis of previously published CLIP-seq data, we noticed that hnRNP A2B1 peaks overlap with RBFOX2 peaks at intron 9 and intron 10 of *PTBP2* (Supplementary Fig. 9a), suggesting that hnRNP A2B1 and RBFOX2 may occupy the same region in the *PTBP2* exon 10-flanking introns. Therefore, we next set out to address whether hnRNP A2B1 affects the RNA-binding ability of SON and/or RBFOX2. Our CLIP with SON antibody and RBFOX2 antibody and subsequent qPCR demonstrated that knockdown of *hnRNP A2B1* markedly reduced SON-binding to *PTBP2* exon 10 splice junctions and the flanking introns (Fig. 5g). Conversely, RBFOX2 binding to those areas was remarkably increased upon *hnRNP A2B1* knockdown (Fig. 5h). Taken together, these results indicated that hnRNP A2B1 is a key component of the SON complex required for SON recruitment to the *PTBP2* exon 10-flanking regions in order to block RBFOX2 recruitment to those areas.

**SON and RBFOX2 competitively regulate alternative splicing of cassette exons in patient-derived GSCs and primary GBM samples.** Based on our finding on the competitive binding of SON and RBFOX2 to the *PTBP2* transcript and their opposite effects on the cassette exon usage, we hypothesize that SON may affect many other RBFOX2-mediated exon inclusion events besides *PTBP2*. For the cassette exons positively regulated (included) by RBFOX2, we selected exon 11 of *INSR* (insulin receptor), exon 4 of *ECT2* (epithelial cell transforming 2)[54,57], exon 23 of *KIF21A*[55], and exon 9 of *CLIP1*[57]. In addition, we found that both RBFOX2 and hnRNP A2B1 peaks are located at the intron sequence downstream of the cassette exon (Supplementary Fig. 9b). Our RT-PCR data demonstrated that *SON* knockdown indeed significantly increases the inclusion of all of these cassette exons in U87MG cells as well as two different GSC lines (Fig. 6a). Furthermore, CLIP-PCR confirmed that SON indeed interacts with these RBFOX2 target transcripts (Fig. 6b). These findings indicate that SON exerts an inhibitory function against RBFOX2-mediated cassette exon inclusion in GBM cells. Using *ECT2* exon 4 as a cassette exon regulated by both RBFOX2 and SON, we further confirmed that dual knockdown of *hnRNP A2B1* and *SON* further enhances *ECT2* exon 4 inclusion, confirming that hnRNP A2B1 works together with SON to maximally repress this cassette exon (Fig. 6c). In contrast, *SON* knockdown-mediated exon 4

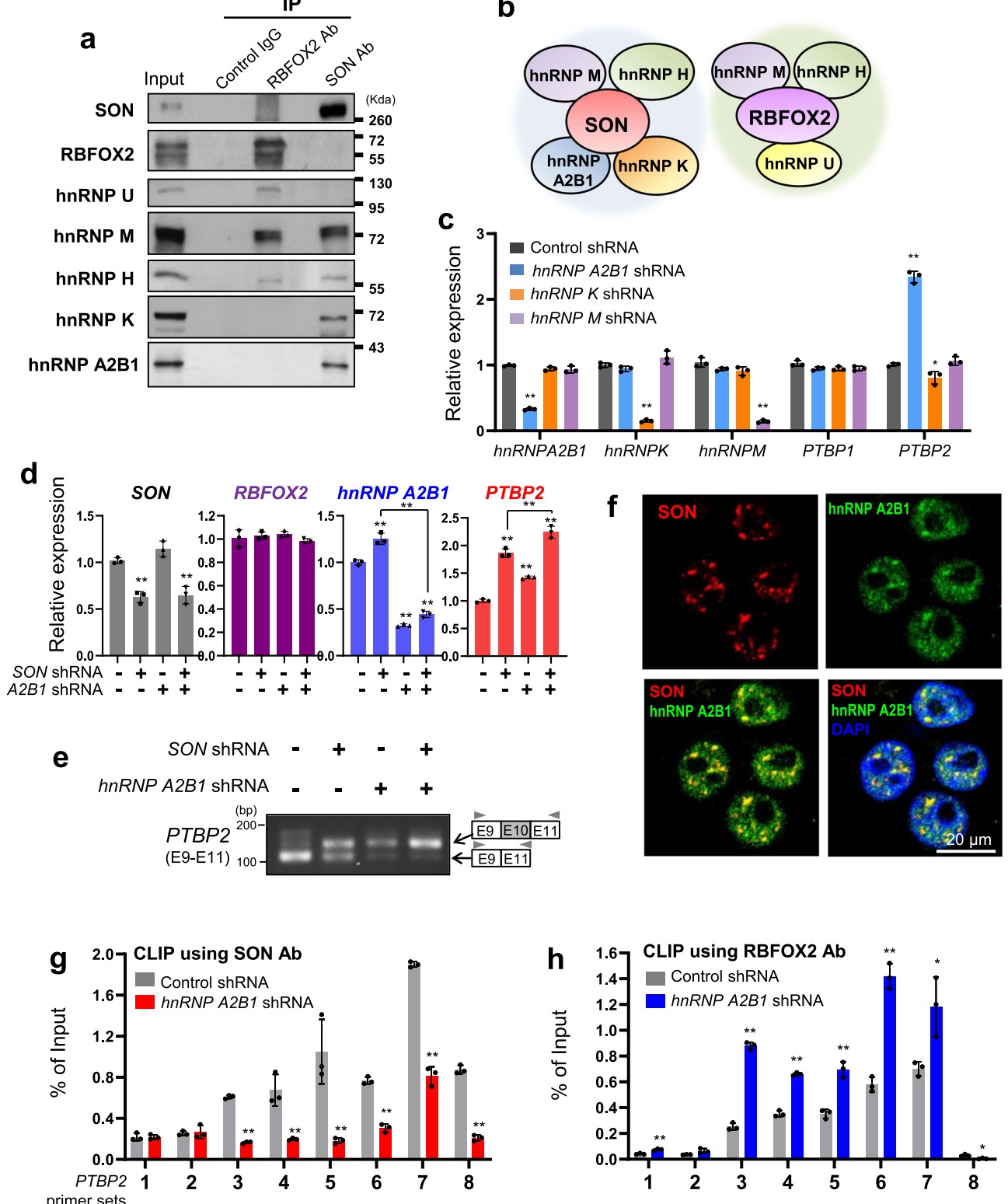

inclusion was not observed in the *RBFOX2* knockdown condition (Fig. 6c), confirming that RBFOX2 action is required to include this cassette exon upon *SON* knockdown.

We next examined whether RBFOX2-mediated cassette exon inclusion is altered in our glioma patient cohort. In addition to SON (Fig. 1b), the hnRNP A2B1 levels were also increased in glioma patient samples (Fig. 6d, e). A positive correlation between SON and hnRNP A2B1 levels was also confirmed from

REMBRANDT, TCGA and CGGA database analyses (Supplementary Fig. 10). However, RBFOX2 expression levels did not show any difference between the normal brain and the malignant glioma patient samples (Fig. 6d, e). We designed specific primers to calculate the ratio of cassette exon skipping and inclusion (Fig. 6f) and RT-qPCR analysis showed that the RBFOX2-targeted cassette exons are largely skipped despite no changes in the RBFOX2 levels in our patent cohort. The ratios of *PTBP2*

**Fig. 5 hnRNP A2B1 interacts with SON, but not with RBFOX2, and facilitates target RNA binding of SON while inhibiting the RBFOX2-RNA interaction.**
**a** Immunoprecipitation (IP) and Western blot analysis confirming the interaction of SON or RBFOX2 with indicated hnRNP proteins. U87MG cell lysates were immunoprecipitated with control IgG, SON antibody, or RBFOX2 antibody and analyzed by Western blotting with the indicated antibodies. Data are representative of $n = 3$ independent experiments. **b** The hnRNP components within the SON complex and the RBFOX2 complex identified from (**a**). **c** RT-qPCR analyses showing the effect of knockdown of three different hnRNPs (*hnRNP A2B1*, *hnRNP K*, and *hnRNP M*) on the expression of *PTBP1* and *PTBP2* in U87MG cells ($n = 3$). The expression levels of indicated hnRNPs were also measured to confirm the knockdown efficiency. **d** RT-qPCR analysis of *SON*, *RBFOX2*, *hnRNP A2B1*, and *PTBP2* in U87MG cells expressing control or *SON* shRNA and infected with or without *hnRNP A2B1* shRNA ($n = 3$). **e** The effects of *SON* knockdown and/or *RBFOX2* knockdown on *PTBP2* exon 10 inclusion were examined by RT-PCR. Data are representative of $n = 3$ independent experiments. **f** Cellular localization of SON (red) and hnRNPA2B1 (green) in U87MG cells were shown by immunostaining. Blue, DAPI. Scale bar: 20 μm. **g**, **h** The effects of *hnRNP A2B1* knockdown on SON or RBFOX2 binding to the *PTBP2* pre-mRNAs were measured by CLIP-qPCR. The locations of the qPCR primer sets indicated as 1–8 are shown in Fig. 4e. The qPCR experiments were performed three times. Error bars in all graphs represent the standard deviation (SD) of tests. NS not significant, $*p < 0.05$, $**p < 0.01$. Statistical significance was determined by an unpaired two-tailed *t*-test. Source data are provided in the Source data file.

exon 10 skipping/inclusion, *INSR* exon 11 skipping/inclusion (the ratio of *INSR-A/INSR-B*) and *ECT2* exon 4 skipping/inclusion are indeed significantly increased in malignant glioma, especially in GBM patient samples when compared to normal brain (Fig. 6g). In addition, analyses of a previously published exon array database[33] confirmed that GBM patients have a high level of *INSR* exon 11 skipping and *ECT2* exon 4 skipping (Fig. 6h). These findings suggest that in GBM patients, although the RBFOX2 expression is not altered, RBFOX2-mediated cassette exon inclusion events are significantly impaired when SON and hnRNP A2B1 are overexpressed due to the competitive action of the SON-hnRNP A2B1 complex blocking RBFOX2 function, revealing a previously unrecognized mechanism repressing RBFOX2 activity in GBM.

**SON knockdown decreases cell growth and clonogenic capacity in GSCs and concurrent inhibition of SON and hnRNP A2B1 potentiates these inhibitory effects.** We next investigated the effect of *SON* knockdown as well as its combinational effects with knockdown of *RBFOX2* and/or *hnRNP A2B1* (Supplementary Fig. 11a) on GBM cell viability and sphere formation. We found that *SON* knockdown significantly reduces the growth rate of patient-derived GSCs in liquid culture (Fig. 7a) and the size and number of spheres in the sphere-forming assay (Fig. 7b, c). Since we found that RBFOX2 expression is required for SON knockdown-mediated PTBP2 upregulation and hnRNP A2B1 facilitates SON recruitment to *PTBP2* transcripts, we further assessed the involvement of RBFOX2 and hnRNP A2B1 in the inhibitory effects on cell proliferation as well as sphere formation associated with *SON* knockdown. First, to assess the role of hnRNP A2B1, we infected control or *SON* shRNA-expressing GSCs with *hnRNP A2B1* shRNA lentivirus. While knockdown of *hnRNP A2B1* alone showed moderate effects on cell proliferation and sphere formation, simultaneous knockdown of both *SON* and *hnRNP A2B1* showed a stronger suppressive effect than *SON* or *hnRNP A2B1* single gene knockdown. Furthermore, the *SON* knockdown-mediated inhibitory effect on cell growth and sphere formation was partially abrogated by depletion of *RBFOX2* (Fig. 7a–c), which further supports our molecular mechanism studies demonstrating a competition between SON and RBFOX2 for alternative splicing.

To address whether these changes are relevant to stemness of GSCs, we performed alkaline phosphatase assays using nitro blue tetrazolium chloride (NBT) staining (Fig. 7d, e). We demonstrated that simultaneous knockdown of *SON* and *hnRNP A2B1* showed a more striking effect on reducing the positively stained spheres compared to control or single gene knockdown. In addition, *RBFOX2* depletion partially abrogated the inhibitory effect of *SON* knockdown on the formation of NBT-positive spheres, demonstrating that *SON* knockdown leads to decreased

stemness and clonogenic capacity of GSCs partially through enhancing the RBFOX2 activity.

These results reveal that reduction of SON has a significant effect on blocking GBM cell growth and cancer stem cell maintenance and the expression status of hnRNP A2B1 and RBFOX2 are critical factors that would affect the outcome of SON inhibition in GBM cells.

**SON knockdown suppresses intracranial tumor growth in vivo.** To examine the effect of *SON* knockdown on GBM tumor formation and growth in vivo, we implanted *SON* shRNA-expressing GSC#83 and control GSC#83 cells (Supplementary Fig. 12a–c) into immunocompromised mice intracranially to generate orthotopic xenografts. We found that, while control GSCs caused a morbid condition with severe neurologic symptoms and weight loss by 11 days post-implant (Supplementary Fig. 12d, e), mice implanted with *SON* shRNA-expressing GSCs did not show those symptoms. Examination of the mouse brain sections revealed that SON-depleted GSCs form significantly smaller tumors compared to control GSCs (Fig. 8a, b). From immunostaining of the tumor areas in the brain sections, we found that both SON expression and PTBP1 expression levels are significantly downregulated in tumors formed by *SON* shRNA-expressing GSCs (Fig. 8c, d), consistent with our in vitro data. Importantly, we also found that SOX2, a well-known GBM cell stemness marker, is almost undetectable in the *SON* shRNA-expressing tumor section while control tumors showed robust SOX2 expression (Fig. 8e). Taken together, our data demonstrated that SON knockdown in GSCs inhibits tumor growth and reduces stemness in vivo.

**Discussion**
Our current knowledge on the basic biology of GBM is limited to a handful of genetic mutations and the mechanism by which non-genetic events contribute to abnormal gene expression in GBM remains largely unknown. Despite the fact that the brain is the organ with the highest level of alternative splicing events, how multiple alternative splicing programs are regulated in GBM has been minimally explored[13]. Here, our study identified that SON is highly expressed in malignant gliomas, especially in GBM, and the high level of SON promotes the removal of detained introns from the PTBP1 transcript, thereby upregulating PTBP1 expression and activating PTBP1-mediated oncogenic splicing program. At the same time, SON antagonizes the RBFOX2-mediated, pro-neuronal differentiation splicing program, such as induction of PTBP2 expression through alternative splicing, by cooperating with hnRNP A2B1. From both in vitro analysis of human GBM samples and in vivo GBM xenograft studies, we provide evidence that reduction of SON triggers multiple changes in PTBP1- and

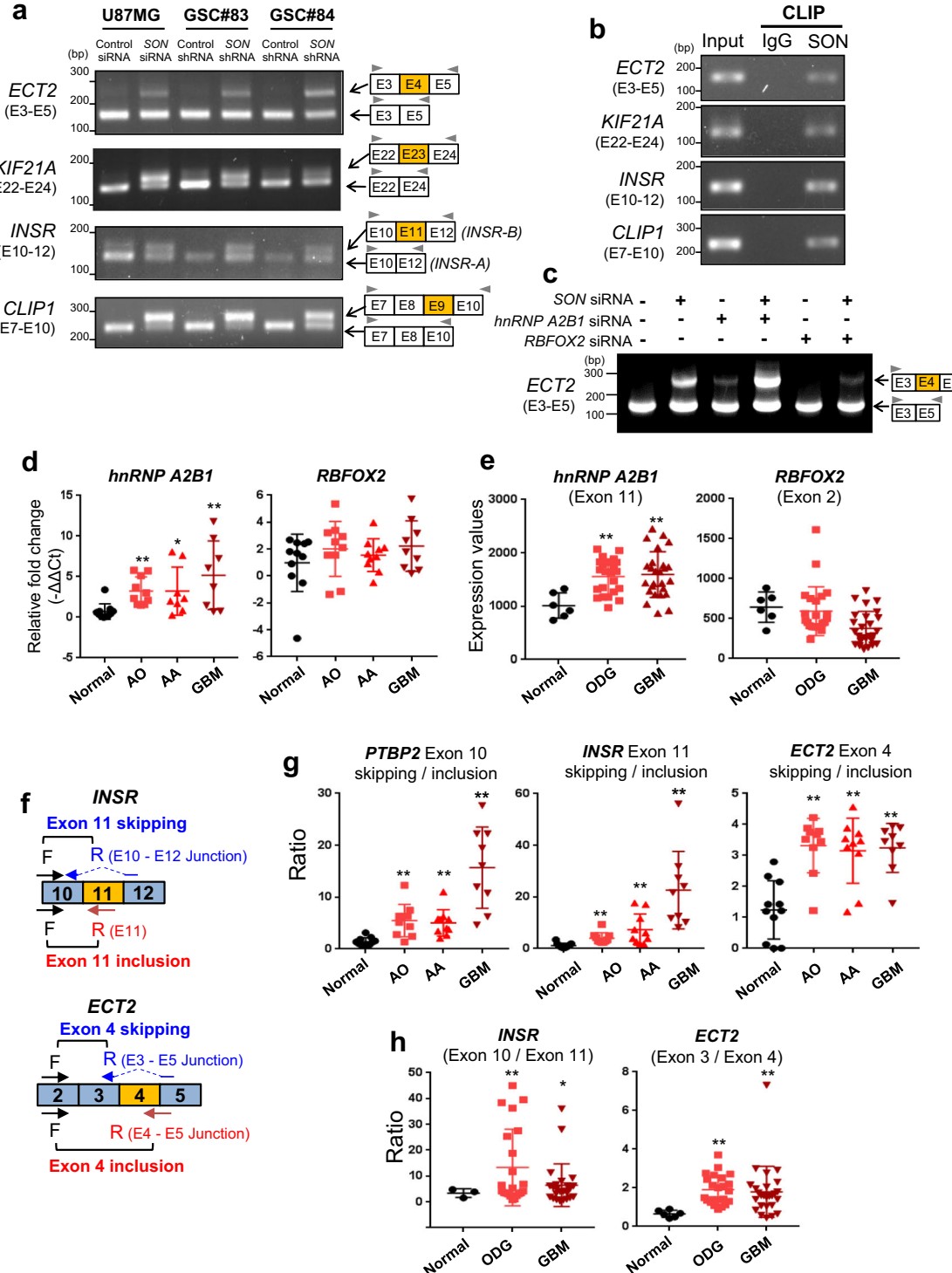

RBFOX2-mediated alternative splicing programs, resulting in a reduction in GBM cell growth and eradicating stemness of GSCs (models in Fig. 9).

Advances in whole-genome and exome sequencing initiated our discoveries of the high frequency of mutations in spliceosome-associated genes, such as *SRSF2*, *SF3B1,* and *U2AF*, particularly, in hematologic malignancies, which has led us to revisit the field of RNA splicing and recognize its significant roles in cancer development and progression[11,12]. Further discoveries of overexpression of several hnRNP proteins have also increased our appreciation of splicing factor functions in cancer[58]. Nevertheless,

information regarding whether cancer-specific changes in splicing factor activity or expression are connected with each other is lacking. Our study discovered SON as a factor positioned at the apex of the splicing factor hierarchy, governing expression and the activity of multiple splicing factors and as well as splicing factor crosstalk in GBM. Our work reveals overexpression of a single splicing factor that is independent of genetic mutations can trigger a series of changes in multiple splicing programs to regulate oncogenesis and neuronal differentiation in GBM.

One of the key findings of our study is the identification of SON as a critical regulator of the PTBP1/PTBP2 switch in GBM.

**Fig. 6 RBFOX2 target cassette exons are frequently skipped in human GBM patients due to the overexpression of SON and hnRNP A2B1 despite no changes in the RBFOX2 level. a** Cassette exon inclusion of RBFOX2 targets, *ECT2, KIF21A, INSR,* and *CLIP1,* is significantly increased upon *SON* depletion in U87MG cells as well as GSCs. Data are representative of $n = 3$ independent experiments. Cassette exons are indicated in orange. **b** CLIP-PCR assay demonstrating the direct interaction between the SON protein and the transcripts of indicated genes containing RBFOX2-targeted cassette exons. Data are representative of $n = 3$ independent experiments. **c** RT-PCR analyses of inclusion and skipping *ECT2* exon 4, an RBFOX2-target exon, under the indicated conditions of siRNA-mediated knockdown of *SON, hnRNP A2B1,* and/or *RBFOX2.* Data are representative of $n = 3$ independent experiments. **d** RT-qPCR assays of *hnRNP A2B1* and *RBFOX2* in malignant glioma patient samples (AO, AA, GBM) and normal brain samples. (Normal, $n = 11$; AO, $n = 10$; AA, $n = 10$; GBM, $n = 9$). **e** *hnRNP A2B1* and *RBFOX2* mRNA expression analyzed using the Exon Expression Arrays dataset. (Normal, $n = 6$; ODG, $n = 23$; GBM, $n = 26$). **f** A schematic illustrating the qPCR strategy and primers used to measure *INSR* exon 11 skipping/inclusion and *ECT2* exon 4 skipping/inclusion. **g** RT-qPCR validation of specific exon-skipped transcripts levels of *PTBP2* (exon 10), *INSR* (exon 11), and *ECT2* (exon 4) in malignant glioma patients samples (AO, AA, GBM) and normal brain samples. (Normal, $n = 11$; AO, $n = 10$; AA, $n = 10$; GBM, $n = 9$). **h** Exon usage ratio of the alternative exon 11 (RBFOX2-targeted cassette exon) and exon 10 (constitutive exon) of *INSR* and exon 4 (RBFOX2-targeted cassette exon) and exon 3 (constitutive exon) of *ECT2* in malignant glioma patients (ODG, GBM) and normal brain samples from the Exon Expression arrays dataset. (Normal, $n = 6$; ODG, n = 23; GBM, $n = 26$). Error bars in all graphs represent the standard deviation (SD) of tests. NS not significant, $*p < 0.05$, $**p < 0.01$. Statistical significance was determined by an unpaired two-tailed *t*-test. Source data are provided in the Source data file.

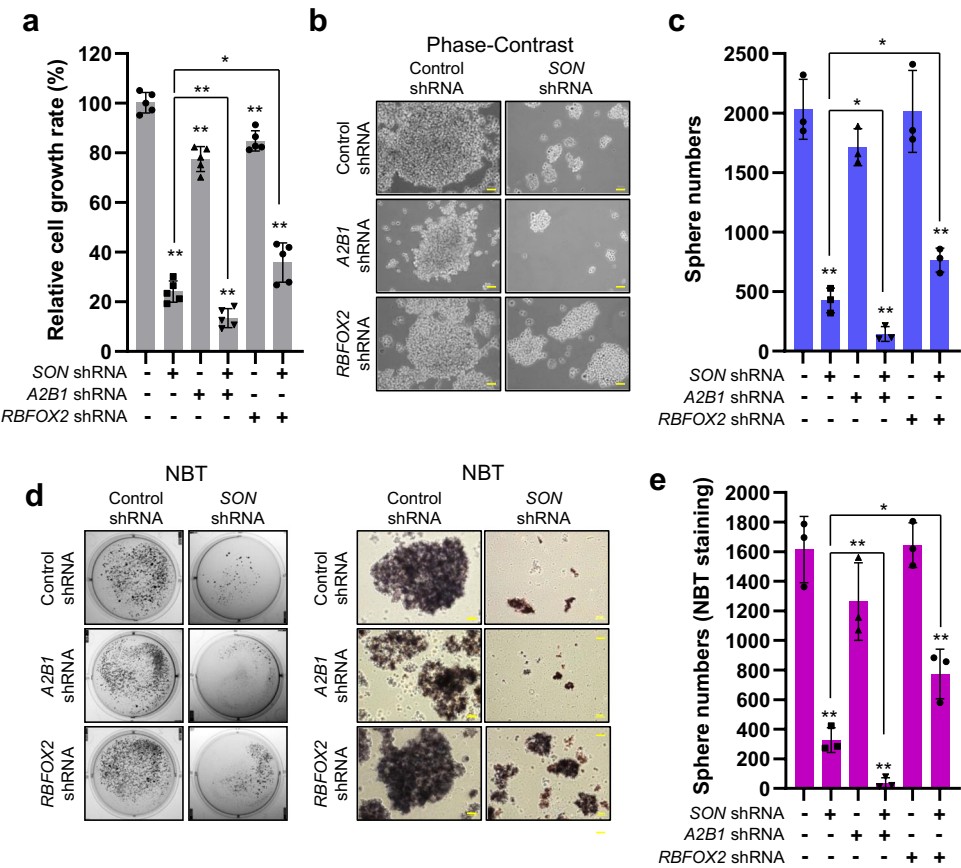

**Fig. 7** *SON* **depletion leads to suppression of proliferation, sphere formation, and clonogenic ability of GSCs, and these effects are facilitated by** *hnRNP A2B1* **knockdown but abrogated by** *RBFOX2* **knockdown. a** Cell growth rate was determined by MTT assay of GSC #83 expressing indicated shRNAs ($n = 5$). **b, c** Representative phase-contrast images of the GSC spheres (**b**) and a graph indicating sphere numbers (**c**, $n = 3$) formed by the GSCs with the knockdown of the indicated genes. Images are representative of $n = 3$ independent experiments. Yellow bars, 50 μm. **d** Alkaline phosphatase assays were conducted by nitro blue tetrazolium chloride (NBT) staining with GSCs to assess the effect of knockdown of the indicated genes in GSC sphere formation. Representative images of NBT-stained spheres ($n = 3$) viewed as a whole plate (left) and the magnified images (right). Yellow bars, 50 μm. **e** The histogram indicates the number of NBT-positive spheres ($n = 3$). Error bars in all graphs represent the standard deviation (SD) of tests. NS not significant, $*p < 0.05$, $**p < 0.01$. Statistical significance was determined by an unpaired two-tailed *t*-test. Source data are provided in the Source data file.

Since PTBP1 and PTBP2 regulate a large number of cassette exons during neuronal differentiation, inhibition of PTBP1 and concurrent upregulation of PTBP2 is required for initiation of neural differentiation[17–19]. PTBP2 upregulation has largely been thought as a result of reduction of its suppressor PTBP1, and few upstream regulators for PTBP1 and PTBP2 have been reported (e.g., miR-124 for PTBP1[45], RBFOX2 for PTBP2[53]). However,

any other master factors that can regulate the PTBP1/PTBP2 switch is completely unknown. We found that SON not only increases PTBP1 expression, but also blocks PTBP2 expression in a PTBP1-independent manner and this regulation occurs at the RNA splicing level, revealing a previously unrecognized RNA splicing cascade that controls a large number of alternative splicing events.

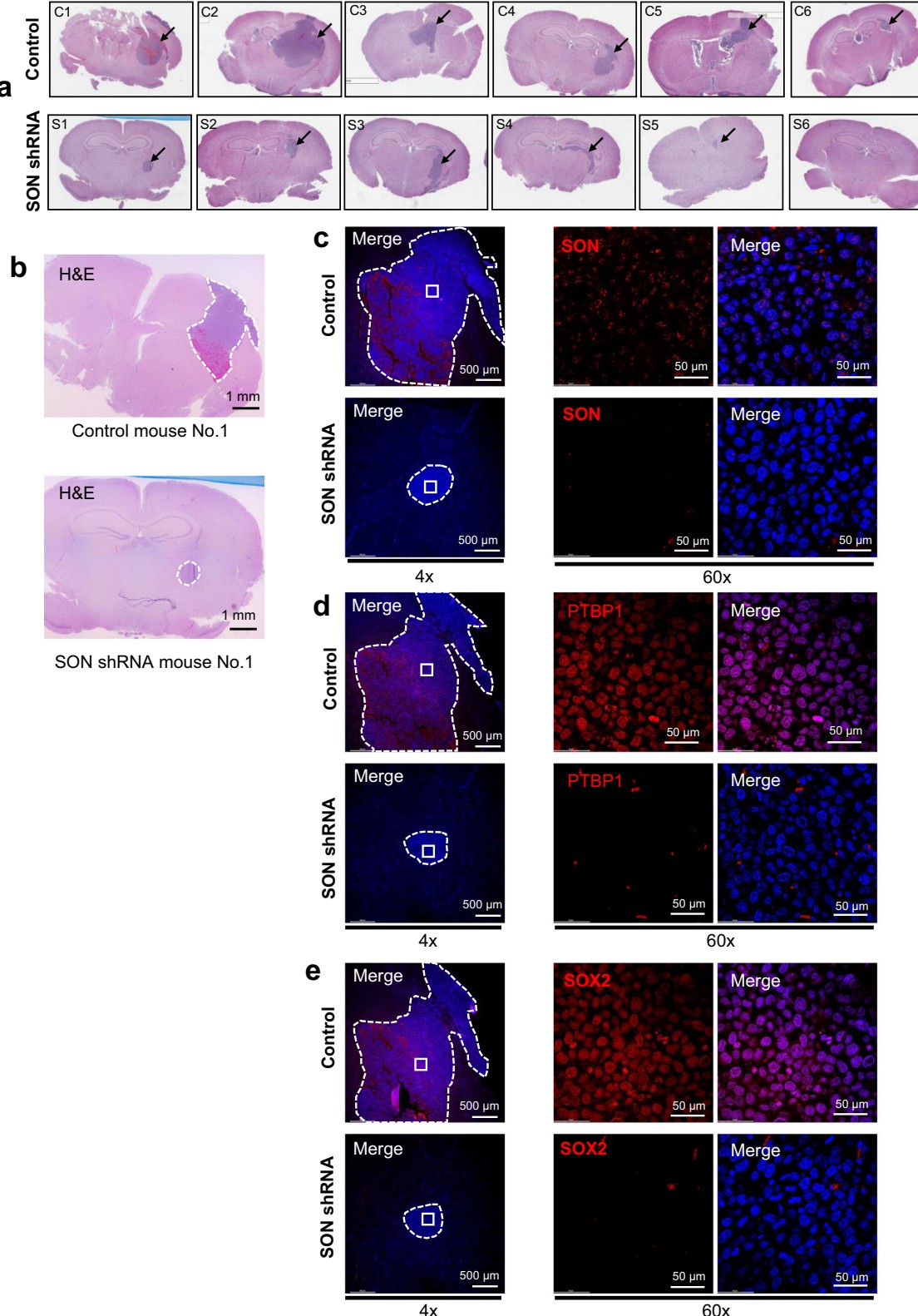

**Fig. 8 SON depletion in GSCs leads to suppression of tumor growth in vivo. a** Representative H&E staining images of mouse brain sections showing tumor growth (black arrows) resulting from stereotaxic injection of GSC#83 cells transduced with control or *SON* shRNA lentivirus (*n* = 6). C1–C6 and S1–S6 indicate individual mice injected with control (C) and *SON* shRNA (S)-transduced GSC#83. **b** H&E sections that were closest to the sections used for immunofluorescence staining in panel (**c**–**e**). Areas marked with dotted lines indicate the tumors. **c**–**e** Representative immunofluorescence staining images of SON (**c**), PTBP1 (**d**), and SOX2 (**e**) (red) together with DAPI (blue) were shown for the tumor formed by GSCs expressing control or *SON* shRNA (*n* = 4). Dotted lines indicate the tumor and the area within the square (marked with while solid lines) was used for 60× imaging shown in (**c**–**e**). Scale bar: H&E: 1 mm, 4×: 500 μm, and 60×: 50 μm.

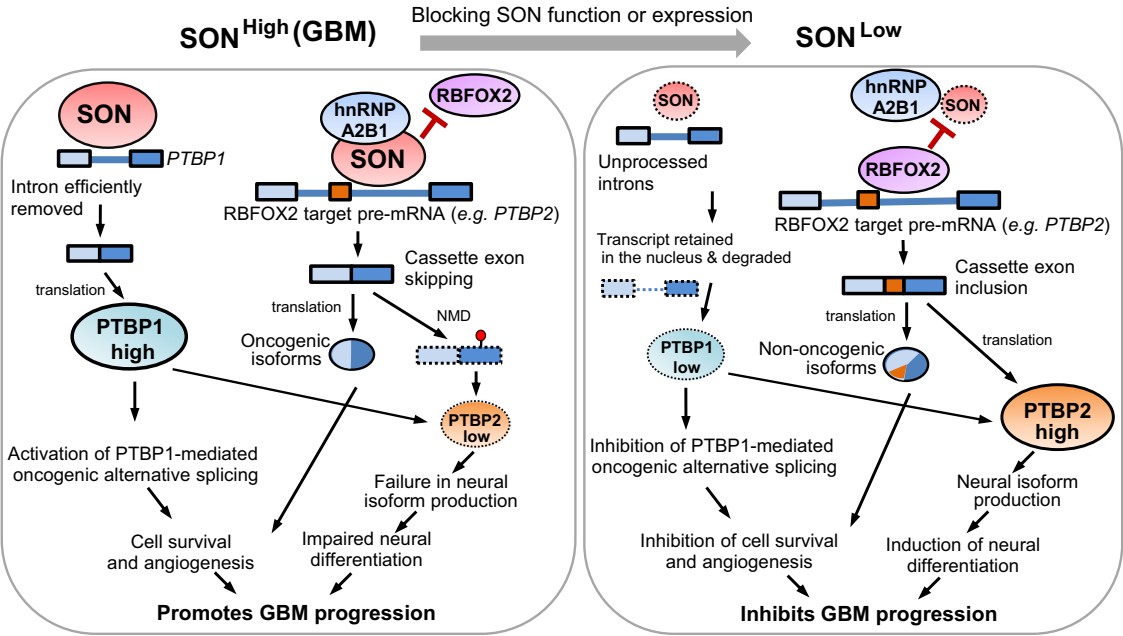

**Fig. 9 Proposed molecular mechanisms of SON action in the regulation of RNA splicing in GBM.** SON upregulates PTBP1 to facilitate the PTBP1-mediated oncogenic splicing program while cooperating with hnRNP A2B1 to antagonize RBFOX2 function, leading to skipping of RBFOX2-activated cassette exons, including the PTBP2 neuronal exon. Therefore, SON triggers splicing programs that promote cell survival while blocking neuronal differentiation. Inhibition of SON function or reducing SON expression would be an effective way to re-shape the splicing programs by reversing the PTBP1/PTBP2 ratio as well as restoring RBFOX2 function, which can turn off oncogenic splicing and turn on neuronal differentiation in GBM.

Our finding of SON as an upstream regulator of PTBP1 expression particularly highlights the clinical significance of SON, since PTBP1 has been recently shown as an oncogenic splicing factor in GBM as well as other multiple cancer types (e.g., in breast cancer, colorectal cancer, and inflammation-associated cancers). PTBP1-mediated cassette exon skipping mainly leads to the production of protein isoforms that promote cell proliferation, angiogenesis, and cell motility[18,21,42,59]. While the oncogenic role of the PTBP1-mediated RNA splicing program has been demonstrated, the mechanism of PTBP1 regulation in cancer cells remains elusive. In addition, two recent reports highlighted the therapeutic value of PTBP1 inhibition, which leads to robust neuronal differentiation, for neurodegenerative diseases[60,61]. Our study discovered that when SON is reduced, *PTBP* transcripts with detained introns remain in the nucleus, and SON is required for the proper removal of those introns. Importantly, induction of intron detention and subsequent gene downregulation has been shown to be an effective way to sensitize glioblastoma cells to TMZ treatment since many of the genes containing detained intron are required for glioblastoma cell proliferation and DNA repair[51]. So far, the factors required for timely removal of detained introns are totally unknown. Our finding on the SON function in intron removal from the PTBP1 transcript suggests potential roles of SON in removing detained introns from many other transcripts, thereby promoting cancer cell growth. A recent study reported that aberrant splicing of BAF45d (a member of the SWI/SNF complex) is mediated by PTBP1, and PTBP1 is also transcriptionally activated by the exon 6A-skipped isoform of BAF45d (*BAF45d/6A−*) in GBM[62]. In our GSCs, we could detect *BAF45d/6A−*, but not the exon 6A-included form (*BAF45d/6A+*) even under the condition of SON and hnRNP A2B1 knockdown (Supplementary Fig. 11b), suggesting that PTBP1 could still be activated transcriptionally. However, we showed that SON knockdown can effectively block PTBP1 mRNA and protein levels under this condition. These findings indicate that PTBP1 can be regulated by multiple steps

(e.g., transcription, microRNA action and RNA splicing) in GBM, and our study defines SON as an important factor regulating PTBP1 at the RNA splicing level.

It is interesting to realize that SON promotes either exon inclusion or skipping in a context-dependent manner. Our study suggests that this is due to SON's ability to associate with numerous hnRNP proteins and form a complex that brings SON to the specific hnRNP-binding sites within specific pre-mRNAs. When hnRNP A2B1 recruits SON to the region near the alternative splice sites where RBFOX2 is also recruited to (e.g., *PTBP2* exon 10-flanking introns), SON is able to hinder RBFOX2-binding to target RNA and diminishes RBFOX2-mediated cassette exon inclusion. Although RBFOX2 has been shown to drive mesenchymal tissue-specific RNA splicing in cancer cell lines[63,64], the role of RBFOX2 in GBM and other malignant gliomas is unclear. Importantly, a recent study demonstrated that RBFOX2 mediates alternative splicing events essential for brain development, suggesting that proper RBFOX2 function is required for neuronal differentiation[65,66]. It has been suggested that RBFOX2 can induce both cassette exon inclusion and skipping, depending on its binding sites near the cassette exon. However, further studies have demonstrated that it is difficult to predict RBFOX2 function in exon inclusion or skipping based on its predicted binding sites, because many cassette exons bear RBFOX2-binding motifs at both upstream and downstream introns and RBFOX2 can be recruited to other sites by interacting with multiple hnRNPs and/or other RNA-binding proteins (LASR complex)[56]. Our analysis on malignant glioma patient samples demonstrate that the level of RBFOX2 expression is not altered in these patients, but, surprisingly, RBFOX2-mediated exon inclusion events are diminished in GBM patients. Our study revealed that SON is able to attenuate RBFOX2-mediated cassette exon inclusion, adding another layer to the regulatory mechanism determining context-dependent RBFOX2 activity.

While traditional cancer studies have predominantly focused on genetic mutations that alter protein function and their

downstream pathways, emerging evidence has indicated that tumors have up to 30% more alternative splicing events than normal samples, indicating critical roles of RNA splicing in cancer-specific gene expression. In addition, alternative splicing can generate tumor-specific neoantigens that can be potentially targeted for immunotherapy[67], underscoring the importance of understanding cancer splicing programs to develop therapeutic strategies. It has been shown that inhibition of PTBP1 and its oncogenic splicing program indeed have therapeutic potential for GBM and several other solid tumors[21,37,68]. Our current study suggests that SON inhibition could have significant benefits as compared to PTBP1 inhibition because blocking SON function not only inhibits PTBP-mediated oncogenic splicing but also leads to activation of the splicing program inducing neuronal differentiation, such as PTBP2-mediated neuronal exon inclusion by a mechanism independent of PTBP1. Our GBM orthotopic xenografts also confirmed that SON reduction in GSCs leads to depletion of PTBP1 as well as the well-known GBM stemness marker SOX2, suggesting targeting SON efficiently suppresses the tumorigenic ability of GSCs and inhibits tumor cells' stemness in vivo. When we continued in vitro culturing of part of the GSCs used for intracranial injection, we observed an accumulation of SON shRNA-expressing GSCs at the G2/M phase of cell cycle 72 and 96 h after shRNA infection, indicating G2/M cell cycle arrest (Supplementary Fig.13). While our current study focused on SON knockdown by siRNA or shRNA, the development of SON inhibitors and assessment of the therapeutic potential of targeting SON-mediated splicing in GBM will open the door to an innovative and unconventional method for treating GBM. In addition, our study reveals the roles of hnRNPA2B1 and RBFOX2 in SON-mediated alternative splicing, suggesting that RBFOX2 and hnRNPA2B1 status should be carefully considered when targeting SON in GBM.

## Methods

**Cell lines**. U87MG cells were maintained in a DMEM medium (Sigma-Aldrich), supplemented with 10% fetal bovine serum (Omega Scientific), 100 U/ml penicillin/streptomycin (Sigma-Aldrich). Glioma stem cells (GSC#83 and GSC#84) were maintained in a DMEM/F12 (Thermo Fisher Scientific) supplemented with B-27 supplement (Thermo Fisher Scientific; 1:50), heparin (Sigma-Aldrich; 5 mg/mL), recombinant human basic FGF (PeproTech; 20 ng/mL), and recombinant human EGF (PeproTech; 20 ng/mL). All cell lines were cultured in a humidified incubator at 5% $CO_2$ and 37 °C.

**Brain tumor patient samples**. The brain tumor tissues from patients (AO; Anaplastic Oligodendrioglioma, AA; Anaplastic Astrocytoma, and GBM; glioblastoma) as well as adjacent brain tissues were obtained from the Brain Tumor Biorepository of the University of Alabama at Birmingham (UAB). All samples were obtained after written informed consent according to the UAB IRB-approved protocols, and encoded with randomly generated numbers to prevent patient identification. Details of the samples were listed in Supplementary Table 1. Details of the patients were listed in Supplementary Table 1. The tissues were embedded in paraffin and the brain sections were cut (5 μm thick) on a microtome for hematoxylin and eosin (H&E) staining and immunofluorescence staining.

**Plasmid construction**. Human RBFOX2 cDNA was generated by PCR amplified of HeLa cDNA using forward primer (F-BamHI-KZ-V5-hRBFOX2) and reverse primer (R-XhoI-hRBFOX2), digested by BamHI and XhoI (New England Biolabs), and ligated into pcDNA3.1 vector. The sequence for the V5 tag was added as indicated in the forward primer. The paired primers for gene-specific shRNA constructs were synthesized, annealed and ligated into the pLenti Lox3.7-EGFP or-puro plasmids cut with HpaI and XhoI restriction enzymes (NEB). The oligonucleotides for making these constructs were provided in Supplementary Table 2. All plasmids were verified by sequencing.

**Lentivirus production**. Recombinant lentiviruses were produced by cotransfecting lentiviral shRNA vectors together with three packaging vectors (pMDLg/pRRE, pRSV-REV, and pVSVG) into HEK 293 FT cells using the PEI. Lentiviral supernatants were collected 48 h after transfection and clarified by filtration before use. Ultracentrifugation was performed for lentivirus concentration with the Optima L-100 XP centrifuge (Beckman) using an SW55TI rotor (Beckman) at 25,000 × g

for 2 h at 20 °C. The supernatant was completely removed and virus pellets were resuspended in PBS.

**siRNA and plasmid transfection**. SON siRNA (GCAUUUGGCCCAUCUGAGAtt[28]) directed against human SON; and negative control siRNA (UAACGACGCGAC-GACGUAAtt) were custom synthesis products by Life Technologies (Silencer Select siRNA). U87MG cells were transfected with 80 pmol of control or SON siRNA using lipofectamine RNAi MAX (Thermo Fisher Scientific) according to manufacturer's instructions. For human PTBP1 or human RBFOX2 overexpression, U87MG and HeLa cells were transiently transfected with 1 μg of plasmids (pCMV-Myc-hPTBP1 and pcDNA3.1-V5 hRBFOX2) using PEI.

**Reverse transcription, quantitative PCR (RT-qPCR), and RT-PCR**. Total RNA was isolated from cell lines and human brain tumor patient samples using RNeasy Mini kit (QIAGEN). Each RNA sample was treated with RNase-free DNase I (QIAGEN). Reverse transcription (RT) was carried out with 0.5–1 μg of total RNA using random hexamer and the SuperScript III First-Strand Synthesis Kit (Thermo Fisher Scientific). Quantitative PCR (qPCR) was performed in triplicate reactions on the CFX Connect Real-Time PCR Detection System (Bio-Rad) or the StepOnePlus Real-Time PCR system (Applied Biosystems) using the iTaq Universal SYBR Green Supermix (Bio-Rad) and standard deviations were calculated. All PCR reactions were performed under the following program: initial denaturation step was 95 °C for 10 min, followed by 40 cycles of denaturation at 95 °C for 10 s, and annealing at 60 °C for 30 s. Gene expression levels were normalized to GAPDH. RT-PCR was performed using primer sets targeting specific exons to see the intron retention or exon inclusion of each transcript by Taq polymerase (Thermo Fisher Scientific). PCR primers were listed in Supplementary Table 2.

**Immunoprecipitation (IP)**. Cells were lysed with lysis buffer (50 mM Tris-HCl, pH 8.0/150 mM NaCl/0.5% NP-40/10% glycerol/Protease Inhibitor Cocktail (Roche)/50 U/ml of benzonase nuclease (Sigma) for 1 h. Lysate was spun 13,000 × g for 30 min at 4 °C and pre-cleared with protein A-Sepharose beads (Life Technologies) for 1 h. Then, the pre-cleared lysates were incubated with rabbit IgG, SON antibody or RBFOX2 antibody and protein A-Sepharose beads at 4 °C overnight. Beads were washed four times with the lysis buffer and eluted by boiling in SDS buffer and subjected to Western blot analyses.

**Western blot analysis**. Following gel electrophoresis and transfer of cell extracts or IP samples onto PVDF membranes (Millipore Sigma) were incubated for 1 h in blocking buffer (5% skim milk in TBST). Membranes were incubated with primary antibodies diluted in TBST for overnight at 4 °C, washed by TBST 30 min, and incubated with appropriate horseradish peroxidase (HRP)-conjugated secondary antibodies (Fisher Scientific). Detection was achieved using ECL (Bio-Rad), x-ray film (Alkali Scientific), and HOPE MicroMax X-ray Processor (ClassicXray). Anti-SON (1:5000 or 1:2,000), anti-PTBP1 (1:3000), anti-PTBP2 (1:2000), anti-Nogo (1:1000), anti-RBFOX2 (1:2000), anti-Actin (1:10,000), anti-CREB (1:3000), anti-GAPDH (1:5000), anti-V5 (1:5000) anti-hnRNP K (1:3000), anti-hnRNP H (1:2000), anti-hnRNP M (1:3000), anti-hnRNP A2B1 (1:2000) and anti-hnRNP U (1:2000) antibodies were used in Western blot analysis.

**Purification of SON-interacting proteins**. Cytoplasmic proteins from the cells were removed by incubating cells in cold hypotonic buffer (10 mM HEPES-KOH, pH 7.9, 1.5 mM $MgCl_2$, 10 mM KCl, 0.5 mM DTT, and 0.1% Triton X-100) and centrifugation. From the pelleted nuclei, nuclear proteins were extracted by homogenization and incubation in high-salt buffer (20 mM HEPES-KOH, pH 7.9, 1.5 mM $MgCl_2$, 420 mM NaCl, 0.5 mM DTT, 0.2 mM EDTA, 25% glycerol and protease inhibitor cocktail). Following centrifugation, the nuclear extract was diluted with 1.8 volume of zero-salt buffer (28 mM Tris-HCl, pH 8.0, 0.28% NP-40 and 0.5 mM DTT) to adjust the final NaCl concentration to 150 mM. The resulting nuclear extracts were pre-cleared with protein A-Sepharose beads (Life Technologies) for 1 h and incubated either with rabbit IgG- or SON antibody-conjugated Sepharose beads at 4 °C overnight on a rotator. The beads were transferred to Mini Bio-Spin Chromatography Columns (Bio-Rad), washed with wash buffer (20 nM Tris-HCl, pH 7.6, 100 mM NaCl, 0.1 mM EDTA and 0.05% Tween-20) for four times and then with equilibration buffer (20 mM Tris-HCl, pH 7.6, 100 mM NaCl) for 8 times. The bound proteins were eluted by adding 500 mM Glycine, pH 2.5 and the elute was immediately neutralized using the solution of 1.5 M Tri-HCl pH.8.8.

**Cellular fractionation and isolation of nuclear and cytoplasmic RNA**. Nuclear and cytoplasmic RNAs were prepared from control or siRNA-transfected U87MG cells using the modified Dignam protocol[69]. In brief, harvested cells were resuspended in buffer A (10 mM HEPES pH 7.9/1.5 mM $MgCl_2$/10 mM KCl/1 mM DTT/0.1% NP-40/Protease Inhibitor Cocktail/RNase OUT) and gently homogenized. The lysates were centrifuged and the supernatants were saved as cytoplasmic fractions. The pellets were washed in buffer A and then used as the nuclear fractions. Both nuclear and cytoplasmic parts were lysed by incubation with TRIzol

reagent (Thermo Fisher Scientific) and the genomic DNA were removed by treatment with DNase I (QIAZEN).

**UV cross-linking and RNA-immunoprecipitation (CLIP).** The siRNA- or shRNA-infected U87MG cells were crosslinked by Stratalinker 2400 (Stratagene) for 10 min at room temperature and were stored at −80 °C until lysis. Subsequently frozen lysates were suspended with 1 ml of lysis buffer (100 mM NaCl/10 mM MgCl₂/30 mM Tris-Cl (pH 7.6)/0.5% Triton/1 mM DTT/Protease inhibitor), homogenized using needle and syringe with 30 gentle strokes. Lysates treated with 10 μL DNase I (Roche) at 37 °C for 5 min at $500 \times g$ in a thermomixer. Samples were placed on ice, treated with 10 μl of diluted RNase I (Thermo Fisher Scientific) solution (1:100 by volume in lysis buffer) and incubated at 37 °C for 5 min at $500 \times i$. Lysates were incubated with 2 μl of RNase OUT (Thermo Fisher Scientific) to quench RNase activity on ice for 5 min and spun at 4 °C on $13,000 \times g$ for 10 min. Supernatants were collected and pre-cleared by 10ul protein A beads 1 h at 4 °C with rotation. Samples were incubated with 5 ug of specific antibody or normal immunoglobulin G (IgG) and magnetic bead protein A (Dynabead Protein A, Thermo Fisher Scientific) at 4 °C for 4 h. Beads were washed sequentially four each with lysis buffer, incubated with 100 μl of DNase I solution (2 μl DNase I in 100 μ lysis buffer) in room temperature for 10 min and washed with lysis buffer 2 more time. Pull-downed RNA were isolated using Trizol.

**Cell proliferation assay.** Cell proliferation was measured using MTT assay. A total of $1 \times 104$ cells in 150 μl medium per well were seed into 96-well plates. After 7 days, 20 μl MTT solutions (Thiazolyl Blue Tetrazolium Bromide; 5 mg/ml; Sigma-Aldrich) were added into each well and incubated for 4 h in the dark. Hundred and fifty microliters of solubilization solution (0.1% NP-40, and 1 N HCl in anhydrous isopropanol) was used to dissolve the formazan grain. The absorbance at 570 nm was measured using SYNERGY 4 Multi Mode Microplate Reader (BioTek).

**Sphere formation assay and alkaline phosphatase assay.** $2 \times 10^5$ cells of GSCs were transduced with recombinant shRNA lentivirus and incubated overnight in the above GSC culture media. After 2 days of culture, cells were plated on 0.5% Noble agar (BD bioscience) coated plates. After 7–10 days, spheres were stained by adding nitroblue tetrazolium chloride (NBT) solution for alkaline phosphatase assay[70] per well and incubating plates overnight at 37 °C. The number of spheres or NBT-positive spheres was quantified as the average and standard deviation of at least triplicate determinations. The average number of spheres and their diameter were analyzed by using Fiji software. Sphere colonies with a diameter below 45 μm were excluded from the analysis.

**Orthotopic implantation of GSCs into mice.** For orthotopic implantation of glioma stem cells (GSCs) into the brains of mice, the following methods were used for lentiviral production and cell transduction. Lentiviral particles were generated by co-transfection of four plasmids into 293-FT cells using the TransIT-X2 Transfection reagent, including the packaging vectors [pMDLg/pRRE, pRSV-REV, and pMD2.g(VSVG)] and the lentiviral vectors for expression of either EGFP control (pLentiLox3.7-EGFP) or SON-specific shRNA (pLentiLox3.7-shSON-EGFP). Lentivirus containing supernatant was collected 48 h after transfection and then passed through 0.45 μM filters to isolate viral particles. Lentivirus particles were further concentrated using a Lenti-X concentrator (Takara Bio, Cat# 631231), as per the manufacturer's instructions. For lentiviral transduction of GSCs, glioma stem cells (GSC#83, $1–2 \times 10^5$) were seeded into 6-well plates with growth medium (1 ml) supplemented with polybrene (final concentration, 8 μg/ml). Concentrated lentiviral particles were immediately added into each well (5 μl) and mixed with the cells. Cells were incubated at 32 °C overnight and then the cell culture medium with lentiviral particles was removed and replaced with a fresh medium. The cells were then incubated at 37 °C. After 48 h, cells were then isolated for orthotopic implantation and for Quantitative Real-Time PCR (qRT-PCR) analysis to confirm the loss of SON expression.

Two days after lentiviral transduction of GSC#83 cells to express SON-shRNA or EGFP, the cells ($1 \times 10^5$) were stereotaxically injected into the right striata of SCID-Beige mice (*scid/scid, beige/beige*, Charles River Laboratories) with the following coordinates: anterior-posterior, +2; medial-lateral, +2; dorsal-ventral, −3 mm from the bregma (six mice per group). All mouse experiments were performed at the University of South Alabama under an Institutional Animal Care and Use Committee (IACUC) approved protocol according to NIH guidelines (Protocol 1649394-3, RWS). All mice were euthanized on the 11th day after injection when neuropathological symptoms developed in brain tumor-bearing animals. The H&E staining was performed by the USA Health Biobank at the University of South Alabama using a standard H&E staining protocol. The images were acquired using a Leica Aperio ScanScope XT Slide Scanner.

**Immunofluorescence staining of tumor sections and brain sections.** The tumor sections from human patients and the brain sections from mouse xenografts were rehydrated and permeabilized in sodium citrate buffer (10 mM, pH 6.0) for 20 min. The sections were incubated with blocking solution containing 3% bovine serum albumin (BSA) and 1% goat serum for 1 h at room temperature. The sections were incubated with primary antibodies overnight at 4 °C. Samples were washed with PBS and incubated with conjugated goat anti-mouse, or anti-rabbit secondary antibodies (1:1000) (Alexa Fluor 594 or 488, Thermo Fisher) for 1 h at room temperature. Species-specific IgG or secondary antibodies were used as negative controls. Slides were mounted using Vectashield Antifade mounting medium (Vector Laboratories), and images were acquired with a confocal microscope at ×4, ×20, or ×60 magnification (Nikon A1R, Nikon). Images were processed using Photoshop CS5.

**CLIP-seq analysis.** CLIP-seq read density files (Supplementary Fig. 4) were generated using IGV tools and were viewed in Integrative Genomics Viewer (IGV) (http://www.broadinstitute.org/igv/). Previously published CLIP-seq data for RBFOX2 (GSE88722), hnRNP K (GSE92089), and hnRNP A2B1 (GSE70061) from NCBI GEO database were analyzed. Enriched CLIP-seq regions at specific exon areas were combined together to generate a unified track consisting of all merged enriched regions.

**Cell cycle analysis.** Three days (72 h) or four days (96 h) after GSC#83 transduction via lentivirus with *SON* shRNA or with a non-targeted control (EGFP), cells ($1 \times 10^6$) were collected, washed, and then fixed with 70% ethanol at −30 °C overnight. The cells were then washed twice with ice-cold PBS and resuspended in 0.5 ml FxCycle™ PI/RNase Staining Solution (cat. #F10797, Thermo Fisher Scientific) and incubated for 30 min at RT. DNA content of the stained cells was analyzed by flow cytometry using a FACS Canto II running Diva Software Version 8.3 (BD Biosciences San Jose, CA). The histogram of the cell cycle distribution was generated from 10,000 events per sample. The percentage of the cells in the G1, S and G2/M phases was calculated using ModFit LT 5.0 software (Verity Software House, Topsham, ME).

**Statistical analysis.** All data points in each graph are mean ± SD. and the n is a biological repeat. Differences were analyzed by Student's t test. P-values <0.05 were considered significant.

**Reporting summary.** Further information on research design is available in the Nature Research Reporting Summary linked to this article.

## Data availability
The brain tumor data used for analyzing gene expression and patient survival are available in the GlioVis (http://gliovis.bioinfo.cnio.es/), R2 database (http://r2.amc.nl), and Oncomine database (https://www.oncomine.org/). Previously published CLIP-Sequencing data and exon array results are accessible at the Gene Expression Omnibus (GEO) repository, under accession number GSE88722, GSE92089, GSE70061, and GSE9385. All the other data are available within the article and its Supplementary Information. Source data are provided with this paper.

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

## Acknowledgements

We thank Dr. Lewis Pannell, Dr. Marie Migaud and Lindsay Schambeau (Mitchell Cancer Institute Mass-spectroscopy Facilities) for providing mass-spectrometry services; Terry Pierce and Dr. Veronica Ramirez Alcantara (University of South Alabama Health Biobank) for the preparation of the H&E slides; Dr. Santanu Dasgupta (University of South Alabama Mitchel Cancer Institute) for the help with slide scanning. This work was supported by the NIH grants (R01CA190688 and R01CA236911 to E.-Y.E.A., R01CA148629 to R.W.S., R01HL136432 to S.-T.S.L.), NIH/NCATS/ Center for Clinical and Translational Science CCTS Pilot Program grant (UL1TR003096-01 to E.-Y.E.A.) and the institutional support from the University of Alabama at Birmingham (UAB) School of Medicine, UAB Department of Pathology, and UAB O'Neal Comprehensive Cancer Center (to E.-Y.E.A.).

## Author contributions

J.-H.K. and E.-Y.E.A. conceived the project. J.-H.K., K. J., J.L., R.W.S., S.-T.S.L., and E.-Y.E.A. designed experiments and interpreted data. J.-H.K, K.J., J.L., J.M.M, L.V., J.K.S., A.R., and J.T. performed the experiments. J.L., G.Y.G., and R.W.S. provided human glioma patient samples and established the GSC lines. G.Y.G. and E.K.F. provided critical discussions on experimental design, data interpretation and manuscript writing. J.-H.K. and E.-Y.E.A. wrote the original draft of the manuscript. J.-H.K., K.J., J.L., R.W.S., S.-T.S.L., and E.-Y.E.A. revised and finalized the manuscript with input from all authors.

## Competing interests

The authors declare no competing interests.
