## [Peer Review File · Nature Communications]

SON drives oncogenic RNA splicing in glioblastoma by regulating PTBP1/PTBP2 switching and RBFOX2 activityReviewers' comments:

Reviewer #1 (Remarks to the Author):

Kim et al. show that expression of the RNA-binding protein (RBP) SON is significantly increased in glioblastoma cells and that this correlates with a poor prognosis in glioblastoma patients. SON regulates splicing of transcripts encoding another RBP, PTBP1, suggesting a possible mechanism for the strong positive correlation between SON and PTBP1 levels in cancer samples. Interestingly, SON knockdown is proposed to stimulate expression of the PTBP1 paralog PTBP2 in a manner independent of the previously characterized crosstalk between PTBP1 and PTBP2. Instead, SON appears to repress splicing of a critical exon in the PTBP2 pre-mRNA by collaborating with hnRNP A2B1 and competing with RBFOX2 for overlapping binding sites in PTBP2 pre-mRNA. The authors show that mechanism involving SON, hnRNP A2B1 and RBFOX2 may control splicing of several additional transcripts. Finally, evidence is provided that proliferation and "stemness" of glioblastoma cells depend on SON and hnRNP A2B1 and that the cytostatic effect caused by downregulation of SON can be partially rescued by RBFOX2 knockdown.

The data are potentially interesting since they suggest that a newly identified RBP network including SON as a key component may play an important part in an aggressive type of cancer associated with a high mortality rate. However, several major limitations of this study must be addressed before it can be considered for publication in a high-impact journal.

Specific comments

1. The main message of the manuscript is that SON is a master regulator of RNA splicing that works via the PTBP1-dependent and hnRNP A2B1/RBFOX2-dependent routes. However, this important point is supported by only a handful of RT-PCR analyses of arbitrarily selected target genes. To strengthen their case, the authors should compare transcriptome-wide splicing effects of SON, PTBP1, hnRNP A2B1 and RBFOX2 knockdowns using RNA sequencing or splicing microarrays.
2. The authors argue that the results presented in Fig 2a-e 'indicate that intron 6 of PTBP1 is a "detained intron" that ... likely functions as a rate-limiting factor for production of fully spliced mRNA'. However, we actually do not know if intron 6 is the only PTBP1 intron regulated by SON. Moreover, SON may control PTBP1 expression level in a manner entirely unrelated to intron detention. The first possibility can be addressed by analyzing detention status of other PTBP1 introns. To confirm that intron 6 is necessary and sufficient for regulation of PTBP1 expression level, the authors should prepare recombinant transcripts containing or lacking this intron and examine their response to SON siRNA or shRNA.
3. The authors propose that SON may work in a PTBP1-independent manner based on the comparison of SON and PTBP1 downregulation kinetics in Fig 3b. The main argument is that PTBP2 is significantly upregulated at the 24 hr timepoint, when SON but not PTBP1 levels are significantly reduced by the SON-specific siRNA. Yet, some downregulation of PTBP1 mRNA is detectable already at 24 hr. Since it is possible that this effect is not statistically significant due to a relatively large standard deviation in their qRT-PCR assay, the authors should reanalyze the time-course data by Western blotting with PTBP1-specific antibodies. Interestingly, Supplementary Fig 3 shows that PTBP1 protein is strongly downregulated by the SON siRNA. How many hours after transfection were the samples in Fig S3 collected?
4. Related to the above comment, siRNAs tend to transfect substantially larger percent of cells in vitro in comparison with plasmids. Hence, it is possible that in the rescue experiment shown in Fig. 3e-g siRNA downregulates PTBP1 expression in most cells but only a small subset of cells overexpress Myc-PTBP1 at a level detectable on the Western blot as overexpression. The authors should estimate the fraction of cells expressing Myc-PTBP1 in this experiment using immunofluorescence or a similar method. Another important question in this regard is whether overexpression of Myc-PTBP1 under these conditions is sufficient to rescue siRNA-induced splicing changes of PTBP1 targets other than PTBP2.
5. All Western blot analyses shown in the manuscript must be repeated at least 3 times and all

changes in band intensities quantified using appropriate software. This is an important concern since Supplementary Fig. 3 claims, for example, that "RBFox2 expression is required for SON knockdown-mediated PTBP2 upregulation". However, it actually appears that SON regulates PTBP2 equally well with and without RBFox2 and only the basal expression of PTBP2 changes depending on RBFox2 levels.

6. I am not convinced that Fig 4 actually proves that RBFox2 regulates PTBP2 mRNA independently of PTBP1. A more likely explanation is that inclusion of PTBP2 exon 10 requires both RBFox2 and PTBP1 downregulation, unlike PBX1 that appears to depend on PTBP1 downregulation only. Incidentally, the authors should specify the time points at which the samples analyzed in Fig 4 (and other Figures) were collected. This information is important to know the extent of PTBP1 downregulation.

7. I assume that the pulldown experiment in Fig 5 was done without an RNase treatment. The authors should comment on this in the text to avoid misleading the reader into thinking that SON is a part of a multisubunit protein complex depicted in Fig 5a. Most of the interactions shown in this diagram might be mediated by RNA. Furthermore, the mass-spec data in Supplementary Table 2 are presented simply as a list of proteins making it difficult to evaluate if the partners of SON were significantly enriched in the pulldown fraction. Was this experiment done only once? How many peptides were detected for each protein?

8. Fig 5e-f: Genetically speaking, the synergistic effect of siRNAs against SON and hnRNP A2B1 indicates that these RBPs regulate PTBP2 splicing and expression independently. The authors should discuss this in light of their model.

9. Fig 6a-b: Do RBFox2 and hnRNP A2B1 regulate these additional SON targets? This should be addressed by treating glioblastoma cells with corresponding siRNA.

10. Fig 7: The authors should quantify RBFox2, hnRNP A2B1 and SON protein expression levels in this experiment using Western blotting to make sure that the synergy between shRNAs against A2B1 and SON and the cell viability rescue by the RBFox2 shRNA are not due to knockdown efficiency changes in cells simultaneously transduced with 2 shRNAs.

Reviewer #3 (Remarks to the Author):

Kim et al. manuscript shows an interesting model of splicing regulation in glioblastoma involving several RNA binding proteins and having SON as a central player. The data is of good quality and overall supports the authors' model. However, in my opinion, a little more needs to be done, having in mind the standards of Nature Communications.

Main points:

- The Rembrandt database is down. The authors should use a different source. The expression and survival analyses they presented are not very convincing. In fact, I checked myself SON expression and impact on survival using TCGA data and got quite different results.
- If possible they should include in Figure 1 immuno-stainings of glioma samples using SON Ab.
- The authors show several CLIP-PCR data. I wonder if it would not be easier and better to conduct a CLIP Seq experiment (and maybe also RNAseq). The data would be ideal to further corroborate their model of antagonism. It is possible that these datasets are already available at ENCODE.
- In Figures 4 and 5, authors need to use a mini gene approach to corroborate the changes in splicing and the regulatory mechanism they are proposing.
- Are the interactions with hnRNPs RNA dependent? A co-localization study will further support these interactions.
- I am not sure why authors decided not to analyze hnRNPH1. In fact, they should compare their dataset to an hnRNPH1 study done in GBM. PMID: 26760575

Reviewer #4 (Remarks to the Author):

The paper entitled "SON drives oncogenic RNA splicing in glioblastoma by regulating PTBP1/PTBP2 switching and RBFOX2 activity" by Kim and coworkers describes the SON-mediated RNA splicing as a GBM vulnerability, implicating SON as a potential therapeutic target in malignant brain tumors. In this paper the authors deal with a very important topic which is alternative splicing and that in the case of malignant brain tumors and specifically, gliomas is vastly unexplored. The authors nicely showed the mechanism of how SON controls PTBP1 and subsequently RBFOX2. They have dedicated a big effort to demonstrate how SON mediates the PTBP1/PTBP2 switch. However, the main flaw that I see in this work is that beyond the molecular mechanism is not that clear how this impact stemness, tumorigenicity and the GBM malignant phenotype in general. Functional, studies demonstrating the reversal of the stem phenotype of the GSC should be performed. Does inhibition of SON results in downregulation of stemness markers (Sox2...), how this affect the cell cycle, death... etc. In vivo studies, with orthotopic GBM models should also help to understand the biological role of SON. It is very important to perform loss of function experiments with (SON) in vivo and in vitro to demonstrate its biological role in the context of GBM.

Minor questions:

Does high SON preferentially associated with a specific GBM molecular signature? Could the author comment on that, also in their series is there any correlation with MGMT methylation promoter?

It has also been described that Baf45d aberrant splicing in GBM (mediated by PTBP1) contributes to the maintenance of an undifferentiated phenotype. Moreover, Baf45d transcriptionally controls PTBP1 expression in GBM (Aldave et al., NeuroOncol 2018), therefore affecting many splicing events. Could the authors comment in the redundant regulation by different proteins of PTBP1?. Baf45d is a member of the swi/snf complex that plays an extremely important role in normal development and also in cancer.

Response to Reviewers

Dear Reviewers,

First of all, we are deeply thankful to the reviewers for their critical, valuable and constructive comments on our initial submission. As most of us did, we had many unexpected obstacles while revising this manuscript due to the COVID19 pandemic. In addition, right before the pandemic began, the lead corresponding author's lab moved to a new institution, and the process of setting up the lab and getting approvals for material transfer and bio-safety/animal protocols at the new institution was extremely challenging due to institutional lockdown as well as personal health issues. Despite these hardships, we persistently made efforts to improve our manuscript. We performed the vast majority of the experiments that the reviewers suggested, and we believe that the new results and modified descriptions have further enlightened the molecular mechanisms, strengthened rigor of the data, and improved the biological/clinical significance of our findings. Below are our point-by-point responses.

Reviewer #1 (Remarks to the Author):

Kim et al. show that expression of the RNA-binding protein (RBP) SON is significantly increased in glioblastoma cells and that this correlates with a poor prognosis in glioblastoma patients. SON regulates splicing of transcripts encoding another RBP, PTBP1, suggesting a possible mechanism for the strong positive correlation between SON and PTBP1 levels in cancer samples. Interestingly, SON knockdown is proposed to stimulate expression of the PTBP1 paralog PTBP2 in a manner independent of the previously characterized crosstalk between PTBP1 and PTBP2. Instead, SON appears to repress splicing of a critical exon in the PTBP2 pre-mRNA by collaborating with hnRNP A2B1 and competing with RBFOX2 for overlapping binding sites in PTBP2 pre-mRNA. The authors show that mechanism involving SON, hnRNP A2B1 and RBFOX2 may control splicing of several additional transcripts. Finally, evidence is provided that proliferation and "stemness" of glioblastoma cells depend on SON and hnRNP A2B1 and that the cytostatic effect caused by downregulation of SON can be partially rescued by RBFOX2 knockdown.

The data are potentially interesting since they suggest that a newly identified RBP network including SON as a key component may play an important part in an aggressive type of cancer associated with a high mortality rate. However, several major limitations of this study must be addressed before it can be considered for publication in a high-impact journal.

Specific comments

1. The main message of the manuscript is that SON is a master regulator of RNA splicing that works via the PTBP1-dependent and hnRNP A2B1/RBFOX2-dependent routes. However, this important point is supported by only a handful of RT-PCR analyses of arbitrarily selected target genes. To strengthen their case, the authors should compare transcriptome-wide splicing effects of SON, PTBP1, hnRNP A2B1 and RBFOX2 knockdowns using RNA sequencing or splicing microarrays.

We appreciate the reviewer for this valuable suggestion. Per reviewer's suggestion, we performed RNA-sequencing after knocking down SON in GSCs, and we have completed genome-wide transcriptome analysis and comprehensive splicing analysis. Indeed, we obtained a great amount of

exciting data revealing SON's function not only in the PTBP-PTBP2 axis and RBFOX2-mediated splicing, but also in alternative splicing of numerous other genes; which are associated with metabolism, inflammatory responses, angiogenesis, DNA repair, spliceosomes, and telomere maintenance. All of these pathways are functionally significant in GBM biology. This revealed that SON potentially affects multiple pathways in GBM.

However, when we tried to present these RNA-seq data in this manuscript, we found that it was an overwhelming task to include all the data in one manuscript. Furthermore, the RNA-seq data gave us numerous novel insights that we cannot adequately convey in one manuscript together with our current molecular mechanism-focused work.

The PTBP1-PTBP2 switch has been considered a key event determining early onset of neuronal differentiation, and its significance has been widely studied. In our current manuscript, we provide a novel molecular mechanism controlling the PTBP1-PTBP2 switch and revealed the interplay among SON, hnRNPA2B1 and RBFOX2, and we believe that the mechanistic insight we showed here is as important as genome-wide profiling of target genes and pathways. For this reason, we feel that it would be better not to combine genome-wide transcriptome/splicing analysis with our current molecular mechanism study. We are currently preparing another manuscript focusing on genome-wide transcriptome and alternative splicing changes caused by SON knockdown in GBM.

2. The authors argue that the results presented in Fig 2a-e 'indicate that intron 6 of PTBP1 is a "detained intron" that ... likely functions as a rate-limiting factor for production of fully spliced mRNA'. However, we actually do not know if intron 6 is the only PTBP1 intron regulated by SON. Moreover, SON may control PTBP1 expression level in a manner entirely unrelated to intron detention. The first possibility can be addressed by analyzing detention status of other PTBP1 introns. To confirm that intron 6 is necessary and sufficient for regulation of PTBP1 expression level, the authors should prepare recombinant transcripts containing or lacking this intron and examine their response to SON siRNA or shRNA.

In our initial study, we examined the splice site score at each splice site and selected a few splice sites for examination. As the reviewer suggested, we further examined the processing of every splice site present within PTBP1 pre-mRNA by PCR, and found two more intron detention events in addition to intron 6 upon SON knockdown; intron 4 and intron 5 (**Fig. 2a**). Other intron removals were found not affected (**Supplementary Fig. 3e**). These findings further confirmed that intron detention occurs as a regulatory step of the *PTBP1* transcript, and the three introns (introns 4, 5 and 6) were regulated by SON. We have modified the manuscript accordingly to present these data and revise our descriptions.

3. The authors propose that SON may work in a PTBP1-independent manner based on the comparison of SON and PTBP1 downregulation kinetics in Fig 3b. The main argument is that PTBP2 is significantly upregulated at the 24 hr timepoint, when SON but not PTBP1 levels are significantly reduced by the SON-specific siRNA. Yet, some downregulation of PTBP1 mRNA is detectable already at 24 hr. Since it is possible that this effect is not statistically significant due to a relatively large standard deviation in their qRT-PCR assay, the authors should reanalyze the time-course data by Western blotting with PTBP1-specific antibodies. Interestingly, Supplementary Fig 3 shows that PTBP1 protein is strongly downregulated by the SON siRNA. How many hours after transfection were the samples in Fig S3 collected?

We agree that careful examination of the kinetics of the PTBP1-PTBP2 change is critical to demonstrate the direct regulation of PTBP2 by SON. While we presented 24h, 48h and 72 h time points after SON siRNA transfection in our initial submission, we examined earlier time points (6h, 12h and 24h) to examine SON, PTBP1 and PTBP2 expression as well as PTBP2 exon 9 inclusion (Fig. 3c).

As the reviewer suggested, we performed PTBP1 Western blot, demonstrating that the PTBP1 protein level was largely unchanged up to 24 hours post-siRNA transfection. However, PTBP2 protein was increased as early as 6 hours after SON siRNA transfection, and more than a 2 fold-increase was observed after 12 hours.

We also clearly observed that PTBP2 exon 9 inclusion is increased at these time points. Certainly, we observed the reduction of the PTBP1 protein at later time points, such as 72 h after SON siRNA transfection, which was shown in Fig 3b and Supplementary Fig 6a (which was Supplementary Fig 3 in our initial submission). Taken together, these additional data confirmed that the increase of PTBP2 exon 9 inclusion and subsequent increase of PTBP2 protein precedes the reduction of PTBP1 protein after SON siRNA transfection. Nevertheless, we certainly acknowledge that PTBP1 reduction upon SON knockdown also contributes to PTBP2 upregulation at later time points. With these new data, we have modified the manuscript to make our point clear.

4. Related to the above comment, siRNAs tend to transfect substantially larger percent of cells in vitro in comparison with plasmids. Hence, it is possible that in the rescue experiment shown in Fig. 3e-g siRNA downregulates PTBP1 expression in most cells but only a small subset of cells overexpress Myc-PTBP1 at a level detectable on the Western blot as overexpression. The authors should estimate the fraction of cells expressing Myc-PTBP1 in this experiment using immunofluorescence or a similar method. Another important question in this regard is whether overexpression of Myc-PTBP1 under these conditions is sufficient to rescue siRNA-induced splicing changes of PTBP1 targets other than PTBP2.

We had the exact same concern when we performed the experiments. Although we did not include the data in our initial manuscript, we indeed assessed the transfection efficiency of Myc-PTBP1 using

immunofluorescence staining with a Myc antibody. We achieved over 90% of transfection efficiency and representative images from two different transfections are shown in **Supplementary Fig. 5**.

To address whether overexpression of Myc-PTBP1 under this condition is sufficient to rescue splicing changes of PTBP1 targets other than PTBP2, we analyzed splicing of *PBX1* (exon 7 inclusion/skipping) and *RTN4* (exon 3 inclusion/skipping). As shown in **Fig 3h**, Myc-PTBP1 transfection completely rescued SON siRNA-induced exon 7 inclusion in *PBX1* splicing. *RTN4* exon 3 skipping was also partially restored by Myc-PTBP1. In contrast to *PBX1* exon 7 and *RTN4* exon 3 skipping, PTBP2 exon 9 skipping induced by SON siRNA was not noticeably rescued by Myc-PTBP1 (**Fig 3h**). These new data support our conclusion that PTBP1 overexpression has a minimal effect on PTBP2 exon 10 inclusion when SON is depleted while splicing of other PTBP1 targets, such as *PBX1* and *RTN4*, was completely or partially rescued.

5. All Western blot analyses shown in the manuscript must be repeated at least 3 times and all changes in band intensities quantified using appropriate software. This is an important concern since Supplementary Fig. 3 claims, for example, that "RBFOX2 expression is required for SON knockdown-mediated PTBP2 upregulation". However, it actually appears that SON regulates PTBP2 equally well with and without RBFOX2 and only the basal expression of PTBP2 changes depending on RBFOX2 levels.

We indeed performed all the Western blots at least 3 times to draw our conclusion and presented representative blots. We measured the band density, calculated the average of relative density levels and indicated them under the representative blots.

6. I am not convinced that Fig 4 actually proves that RBFOX2 regulates PTBP2 mRNA independently of PTBP1. A more likely explanation is that inclusion of PTBP2 exon 10 requires both RBFOX2 and PTBP1 downregulation, unlike *PBX1* that appears to depend on PTBP1 downregulation only. Incidentally, the authors should specify the time points at which the samples analyzed in Fig 4 (and other Figures) were collected. This information is important to know the extent of PTBP1 downregulation.

We thank the reviewer's insight into this matter. We realized that we did not describe our experimental condition sufficiently. For all the experiments that employ both RBFOX2 and SON knockdown shown in Figure 4, we first knocked down RBFOX2 using a lentiviral shRNA construct (Lentivirus infection: Day 1) to create RBFOX2-low condition. Then, on the next day, SON knockdown was performed by siRNA transfection (siRNA transfection: Day 2). Two days after SON siRNA transfection (= 3 days after RBFOX2 lentivirus infection), cells were collected for analyses. The expression levels of SON, RBFOX2, PTBP1 and PTBP2 resulting from this procedure are presented in **Figure 4a** (RT-qPCR) and **Figure 4b** (Western blot).

The reason we knockdown RBFOX2 is to evaluate whether RBFOX2 is a critical “mediator” of SON siRNA-mediated PTBP2 exon 9 inclusion. Under this experimental condition, we found that while PTBP1 levels are reduced to a level similar to after single SON knockdown or SON/RBFOX2 double knockdown, PTBP2 upregulation by SON siRNA was much effective when RBFOX2 was present, which suggests RBFOX2 is an important downstream effector (mediator) molecule in the process of SON knockdown-induced PTBP2 regulation.

We clarify that we are not excluding PTBP1 as an important regulator of PTBP2 as PTBP1 reduction was still able to upregulate PTBP2 even in the absence of RBFOX2. Our experiments were to propose an additional mechanism of PTBP2 regulation, which involves SON-RBOX2. We modified our statements in the text to convey our conclusion.

7. I assume that the pulldown experiment in Fig 5 was done without an RNase treatment. The authors should comment on this in the text to avoid misleading the reader into thinking that SON is a part of a multisubunit protein complex depicted in Fig 5a. Most of the interactions shown in this diagram might be mediated by RNA. Furthermore, the mass-spec data in Supplementary Table 2 are presented simply as a list of proteins making it difficult to evaluate if the partners of SON were significantly enriched in the pulldown fraction. Was this experiment done only once? How many peptides were detected for each protein?

As described in the Method section, we performed all the pulldown experiment (Fig 5) after treating the lysate with **Benzonase** (Sigma, E1014-5KU) which digests both DNA and RNA. Therefore, we are confident that the interactions identified here are not mediated by DNA or RNA.

We also included the mass-spec data containing Mascot score, sequence coverage, and the number of peptide identified. We performed the pulldown under extremely harsh conditions and extensive wash steps. Since the number of trypsin-digestible sites sometimes depends on the amino acid composition of the protein (we only detected 4 peptides for SON), we evaluate Mascot Score instead of simply looking at the number of peptides identified. The Mascot score is the summed score for the individual peptides, e.g. peptide masses and peptide fragment ion masses for all peptides matching a given protein. We included the information in **Supplementary Table 2**.

The proteins with minimum Mascot score 43 (corresponding to a peptide false-discovery rate of 0.09% based on comparison to the random database) were considered as potential partners (The Mascot score is $-10 \times \log(P)$, where P is the probability that the observed match is a random event). Although we performed an unbiased mass-spec experiment once to select candidates, we repeated IP and WB for 3 times to verify different hnRNP-binding profiles of SON and RBFOX2, and to confirm the SON-hnRNP A2B1 interaction. In this revised manuscript, we presented the diagram of SON-interacting proteins in Supplementary Fig 7 to present potential candidates and the verified result in Fig 5a with a representative IP-WB result.

8. Fig 5e-f: Genetically speaking, the synergistic effect of siRNAs against SON and hnRNP A2B1 indicates that these RBPs regulate PTBP2 splicing and expression independently. The authors should discuss this in light of their model.

We thank the reviewer for the insight and we agree that SON and hnRNP A2B1 can suppress PTBP2 exon 10 inclusion independent of each other. At the same time, we also observed that SON recruitment to the PTBP2 transcript is enhanced by hnRNP A2B1, indicating their cooperation. Based on these data, we described that SON and hnRNP A2B1 function cooperatively as well as independently (In the result section pertaining to Figure 5).

9. Fig 6a-b: Do RBFOX2 and hnRNP A2B1 regulate these additional SON targets? This should be addressed by treating glioblastoma cells with corresponding siRNA.

In addition to PTBP2, we examined how SON, RBFOX2 and hnRNP A2B1 regulate inclusion/skipping of *ECT2* exon 4, a cassette exon known to be an RBFOX2 target. Using different combinations of the siRNAs for SON, hnRNP A2B1 or RBFOX2, we confirmed that both SON knockdown (SON siRNA) and hnRNP A2B1 knockdown (hnRNP A2B1 siRNA) lead to *ECT2* exon 4 inclusion, and dual knockdown of *hnRNP A2B1* and *SON* further enhances *ECT2* exon 4 inclusion, confirming that knocking down both hnRNP A2B1 and SON can maximally repress this cassette exon. In contrast, when RBFOX2 was depleted (RBFOX2 siRNA), SON knockdown-mediated exon 4 inclusion was not efficient, confirming that the RBFOX2 action is required to include this cassette exon upon SON knockdown. We include these data in **Fig 6c** and revised the text accordingly.

10. Fig 7: The authors should quantify RBFOX2, hnRNP A2B1 and SON protein expression levels in this experiment using Western blotting to make sure that the synergy between shRNAs against A2B1 and SON and the cell viability rescue by the RBFOX2 shRNA are not due to knockdown efficiency changes in cells simultaneously transduced with 2 shRNAs.

When we performed the experiments for Figure 7, we collected cells and measured RBFOX2, hnRNP A2B1 and SON expression levels by qPCR. The results confirmed that the simultaneous transduction of two shRNAs not affect the knockdown efficiency. The data are now included in Supplementary Fig 11a.

Reviewer #3 (Remarks to the Author):

Kim et al. manuscript shows an interesting model of splicing regulation in glioblastoma involving several RNA binding proteins and having SON as a central player. The data is of good quality and overall supports the authors' model. However, in my opinion, a little more needs to be done, having in mind the standards of Nature Communications.

Main points:

- The Rembrandt database is down. The authors should use a different source. The expression and survival analyses they presented are not very convincing. In fact, I checked myself SON expression and impact on survival using TCGA data and got quite different results.

When we obtained the survival curves that we included in the initial submission, the old REMBRANDT site was still working. We found that the old site is not available any more. However, the REMBRANDT data sets are available from other data portals.

To revise our manuscript, we completely re-analyzed SON expression and patient survival using Gliovis (<http://gliovis.bioinfo.cnio.es>), a recently established data portal for brain tumor expression datasets. It is important to note that simple classification using the “median” value of a predictor is a frequent problem in medical research. We found that the negative effect of SON expression on patient survival was not as significant if one used “median” as a cutoff value to divide SON-high and SON-low patient groups.

Gliovis data portal provides the tool to identify the “optimal cutoff” value based on “maximally selected rank statistics” (**Supplementary Fig 2a**). This method allows the evaluation of cutoff points, which

provide the classification of observations into two groups by a continuous or ordinal predictor variable. In the new **Figure 1d**, we presented survival curves in which SON-high and SON-low patient groups (from REMBRANDT All_Glioma) are divided by “median” or “Optimal cutoff value”, demonstrating that the impact of SON level on patient survival is more significant when we use optimal cutoff as a cutoff point. In new **Supplementary Fig 2c**, we also showed how the survival curves of GBM patients (REMBRANDT_GBM_All) could be different when “median”, “upper quartile” or “optimal cutoff” were used as a cutoff point of SON level. Using optimal cutoff values, we also analyzed patient survival from TCGA_GBM and CGGA_All Glioma cohorts (**Supplementary Fig 2**). We hope that now these data clearly demonstrate the impact of SON on patient survival.

-If possible they should include in Figure 1 immuno-stainings of glioma samples using SON Ab.

We agreed that immunostaining of glioma patient samples is a valuable approach to evaluate the SON expression level. We stained normal brain and glioblastoma patient samples using SON antibody. The results showed a marked difference in SON levels; the SON staining signal was much stronger in the glioblastoma sample.

Furthermore, we stained these samples with PTBP1 and PTBP2 antibodies, and the results indeed are consistent with our database analysis and qPCR. We included the data in Figure 1 (**Fig 1h, i**).

-The authors show several CLIP-PCR data. I wonder if it would not be easier and better to conduct a CLIP Seq experiment (and maybe also RNAseq). The data would be ideal to further corroborate their model of antagonism. It is possible that these datasets are already available at ENCODE.

While RBFOX2 CLIP-seq data are published and available from the database, SON CLIP-seq has not been done by any other research group. The commercially available SON antibodies are not suitable for CLIP, and we have developed our own SON antibody which works well for CLIP. While we agree that CLIP-seq would be the ultimate way to show the antagonism between SON and RBFOX2, in this manuscript, we focused on the novel function of SON in regulating PTBP1/PTBP1 switch. We recently performed SON CLIP-seq with our own antibody and identified exciting features of transcriptome-wide SON-RNA interactions. However, we feel that it is difficult to include our SON CLIP-seq data in this manuscript because we found not only PTBP1-PTBP2, but also numerous other important RNA targets of SON in our transcriptome-wide analysis of SON-binding sites. We hope that we can report our extensive analyses of transcriptome-wide analyses of SON- and RBFOX2-binding sites in our next manuscript in the near future.

-In Figures 4 and 5, authors need to use a mini gene approach to corroborate the changes in splicing and the regulatory mechanism they are proposing.

As the reviewer suggested, we constructed a minigene covering the PTBP2 exon 9-10-11 region (WT), and minigene splicing analysis demonstrated that SON knockdown indeed remarkably enhances exon 10 inclusion.

We also constructed minigenes with the potential RBFOX2 sites mutated (Mut#1, Mu#2 and Mut#1+2), and the effect of SON knockdown on exon 10 inclusion is significantly diminished in these mutant minigenes. These data further support our finding that RBFOX2's action is required for cassette exon inclusion in the condition of SON knockdown. We included new figures (Fig 4i, j and Supplementary Fig 6b, c) and revised the manuscript accordingly.

-Are the interactions with hnRNPs RNA dependent? A co-localization study will further support these interactions.

As we indicated in our response to Reviewer #1, we treated the cell lysate with Benzonase (which digests DNA and RNA) before we performed protein pulldown assays. Therefore, we are sure that the interactions with hnRNPs we presented here are not RNA-dependent.

To further support SON interaction with hnRNP A2B1, we performed immunostaining for SON and hnRNP A2B1 in U87MG cells. We detected co-localization of SON and hnRNP A2B1 in nuclear speckles, but the localizations of these two proteins were not completely identical (Fig 5f). This result supports the notion that SON and hnRNP A2B1 function cooperatively as well as independently. We propose that the interaction and cooperative action between SON and hnRNP A2B1 occurs at the sites where co-localizations are detected. The results were presented in **Figure 5f**.

-I am not sure why authors decided not to analyze hnRNPH1. In fact, they should compare their dataset to an hnRNPH1 study done in GBM. PMID: 26760575

We appreciate the reviewer's suggestion on hnRNP H1 and the associated reference (PMID: 26760575; Uren et al. 2016, RNA Biology). Uren et al. performed iCLIP and RIP-seq and listed the hnRNP H1-interacting RNAs identified by both methods. We found that PTBP2 was not considered as a "true" target because it was detected only one of two replicates of iCLIP and the p-value of RIP was not considered significant ($p=1$). Therefore, it is likely that hnRNP H1 does not directly regulate PTBP2 splicing.

Nevertheless, we used hnRNP H1 siRNA to examine whether hnRNP H1 indirectly regulates PTBP2 or cooperates with SON. We found that knockdown of hnRNP H1 did not alter either PTBP1 or PTBP2 expression (**Supplementary Fig 8**). Therefore, we concluded that hnRNP H1 does not play a significant role in SON regulation of PTBP1 and PTBP2.

Reviewer #4 (Remarks to the Author):

The paper entitled "SON drives oncogenic RNA splicing in glioblastoma by regulating PTBP1/PTBP2 switching and RBFOX2 activity" by Kim and coworkers describes the SON-mediated RNA splicing as a GBM vulnerability, implicating SON as a potential therapeutic target in malignant brain tumors. In this paper the authors deals with a very important topic which is alternative splicing and that in the case of malignant brain tumors and specifically, gliomas is vastly unexplored. The authors nicely showed the mechanism of how SON controls PTBP1 and subsequently RBFOX2.

They have dedicated a big effort to demonstrated how SON mediates the PTBP1/PTBP2 switch, However, the main flaw that I see in this work is that beyond the molecular mechanism is not that

clear how this impact stemness, tumorigenicity and the GBM malignant phenotype in general. Functional, studies demonstrating the reversal of the stem phenotype of the GSC should be performed. Does inhibition of SON results in downregulation of stemness markers (Sox2...), how this affect the cell cycle, death... etc. In vivo studies, with orthotopic GBM models should also help to understand the biological role of SON. It is very important to perform loss of function experiments with (SON) in vivo and in vitro to demonstrate its biological role in the context of GBM.

We thank the reviewer for the enthusiasm for the molecular mechanism we presented and the valuable suggestions regarding SON's biological function. We performed *in vivo* experiments with mouse orthotopic xenograft models of GBM. We found SON-depleted GSCs formed significantly smaller tumors compared to control GSCs when intracranially injected into mice.

From immunostaining of mouse brains bearing the xenografts, we also found that SOX2, a well-known GBM stemness marker, is almost undetectable in the SON-depleted tumor sections while control tumors showed robust SOX2 expression. In addition, PTBP1 is indeed downregulated in SON-depleted tumor sections. We also analyzed cell cycle in GSCs expressing SON shRNA, which indicated that depletion of SON resulted in a G2/M arrest.

We included these new data in **Fig 8** and **Supplementary Figs 12 and 13**, and revised the manuscript accordingly. We believe that these new data from *in vivo* studies highlighted the biological significance of SON and significantly improved our manuscript.

Minor questions:

Does high SON preferentially associated with a specific GBM molecular signature? Could the author comment on that, also in their series is there any correlation with MGMT methylation promoter?

We thank the reviewer for bringing up this interesting aspect. In this revised manuscript, we analyzed SON expression in different molecular subtypes of GBM (classical, mesenchymal and proneural) using TCGA and REMBRANDT GBM datasets. We also found that SON expression levels are neither significantly different among GBM subtypes nor different G-CIMP (glioma cytosine-phosphate-guanine (CpG) island methylator phenotype) status. We presented these data in Supplementary Fig 1d.

It has also been described that Baf45d aberrant splicing in GBM (mediated by PTBP1) contributes to the maintenance of an indifereciated phenotype. Moreover, Baf45d transcriptionally controls PTBP1 expression in GBM (Aldave et al., NeuroOncol 2018), therefore affecting many splicing events. Could the authors comment in the redundant regulation by different proteins of PTBP1?. Baf45d is a member of the swi/snf complex that plays an extremely important role in normal development and also in cancer.

We thank the reviewer for this suggestion. The paper suggested by the reviewer demonstrated that aberrant splicing of BAF45d (a member of the SWI/SNF complex) is mediated by PTBP1, and PTBP1 is also transcriptionally activated by the exon 6A-skipped isoform of BAF45d (BAF45d/6A-) in GBMs. In our GSCs, we could detect only BAF45d/6A-, but not the exon 6A-included form (BAF45d/6A+) even under the condition of SON and hnRNP A2B1 knockdown (**Supplementary Fig. 11b**), suggesting that PTBP1 could still be activated transcriptionally by BAF45d/6A- when SON is depleted. However, we showed that SON knockdown can effectively block PTBP1 mRNA and protein levels under this condition. These findings indicate that PTBP1 can be regulated by multiple steps (e.g. transcription, microRNA action and RNA splicing) in GBM, and our study defines SON as a novel factor regulating PTBP1 at the RNA splicing level. We revised the Discussion section to include these descriptions.

REVIEWERS' COMMENTS

Reviewer #1 (Remarks to the Author):

The authors have addressed the key concerns of this and other reviewers. I think the manuscript is now suitable for publication.

Reviewer #3 (Remarks to the Author):

Authors included a substantial number of new experiments and analyses and made a great effort to address reviewers' concerns. The new version is a much better article that, in my opinion, merits publication in Nature Communications.

Reviewer #4 (Remarks to the Author):

They authors have done an important effort to address this reviewers concerns

Response to Reviewers

We are very pleased to hear positive comments from the reviewers on our revised manuscript. We sincerely thank the reviewers for their time, effort and insight.

Reviewer #1 (Remarks to the Author):

The authors have addressed the key concerns of this and other reviewers. I think the manuscript is now suitable for publication.

The reviewer previously provided valuable points and insight that greatly strengthened the mechanistic aspect of our manuscript. We sincerely thank the reviewer for the positive comments on our revised manuscript.

Reviewer #3 (Remarks to the Author):

Authors included a substantial number of new experiments and analyses and made a great effort to address reviewers' concerns. The new version is a much better article that, in my opinion, merits publication in Nature Communications.

We thank the reviewer again for the valuable suggestions on GBM patient survival data analysis and the minigene approach. We are glad to hear the reviewer's positive decision.

Reviewer #4 (Remarks to the Author):

They authors have done an important effort to address this reviewers concerns

We performed the *in vivo* experiments per the reviewer's suggestion and the results truly enhanced the biological significance of our work. We thank the reviewer for the positive final decision.